**EMBO** *reports*

# IL-6 promotes tumor growth through immune evasion but is dispensable for cachexia

Young-Yon Kwon [ID] & Sheng Hui [ID] ✉

## Abstract

**Various cytokines have been implicated in cancer cachexia. One such cytokine is IL-6, deemed as a key cachectic factor in mice inoculated with colon carcinoma 26 (C26) cells, a widely used cancer cachexia model. Here we tested the causal role of IL-6 in cancer cachexia by knocking out the *IL-6* gene in C26 cells. We found that the growth of IL-6 KO tumors was dramatically delayed. More strikingly, while IL-6 KO tumors eventually reached the similar size as wild-type tumors, cachexia still took place, despite no elevation in circulating IL-6. In addition, the knockout of leukemia inhibitory factor (LIF), another IL-6 family cytokine proposed as a cachectic factor in the model, also affected tumor growth but not cachexia. We further showed an increase in the infiltration of immune cell population in the IL-6 KO tumors compared with wild-type controls and the defective IL-6 KO tumor growth was rescued in immunodeficient mice while cachexia was not. Thus, IL-6 promotes tumor growth by facilitating immune evasion but is dispensable for cachexia.**

**Keywords** Colon Carcinoma 26; Cancer Cachexia; IL-6; LIF; Tumor Immune Evasion
**Subject Categories** Cancer; Immunology

## Introduction

Cancer cachexia is a systemic wasting syndrome prevalent in cancer patients, contributing to physical disability, poor quality of life, and death (Law, 2022; Lim et al, 2020; Ni and Zhang, 2020; von Haehling et al, 2016). In search for mediators of cancer cachexia, researchers have proposed many cachectic factors over the years, with more than 20 of them listed in recent reviews (Baracos et al, 2018; Ferrer et al, 2023). Targeting these factors, however, has so far not yielded an effective treatment. While this lack of success in translation may arise from differences between preclinical models and humans, it is possible that the real biological roles of these factors have not been sufficiently established in preclinical models.

More than half of the proposed cachectic factors are cytokines (Baracos et al, 2018), including well-known examples such as TNF-α (Beutler and Cerami, 1986; Oliff et al, 1987; Spiegelman and Hotamisligil, 1993), IL-1 (Gelin et al, 1991; Moldawer et al, 1987), IFN-γ (Matthys et al, 1991a; Matthys et al, 1991b) and IL-6 (Matsumoto et al, 1999; Strassmann et al, 1992). Among these, IL-6 is one of the most frequently cited cachectic factors. Its cachectic role was initially proposed in the syngeneic tumor model of colon 26 carcinoma (C26) (Strassmann et al, 1992) and later implicated in various cancer models including genetically engineered mouse models such as the *Apc^{Min/+}* colorectal cancer model and pancreatic ductal adenocarcinoma model (Baltgalvis et al, 2008; Flint et al, 2016) and patient-derived tumor xenograft models such as the HT1080 fibrosarcoma model, TOV21G ovarian cancer model, and A2058 melanoma model (Bernardo et al, 2020; Pettersen et al, 2017; Pettersen et al, 2020). Given the prominent status of IL-6 in cancer cachexia research, here we aimed to closely examine its causal role in cancer cachexia. For this, we focused on the C26 model, where IL-6 was first proposed to be a cachectic factor (Matsumoto et al, 1999; Strassmann et al, 1992), and its cachectic role seems to be best established.

Existing evidence indicates that IL-6 is unlikely a sufficient factor for causing cachexia in the C26 model. Although circulating levels of IL-6 are dramatically increased in mice inoculated with cachectic C26 cells, similar levels of IL-6 have also been measured in mice inoculated with a non-cachectic subclone of C26 cells (Fujimoto-Ouchi et al, 1995; Soda et al, 1994), indicating that IL-6 alone cannot induce cachexia. Furthermore, infusion of IL-6 to healthy and non-cachectic C26 tumor-bearing mice failed to induce body weight loss (Soda et al, 1995; Yasumoto et al, 1995). Although IL-6 infusion to mice injected with CHO cells led to weight loss and muscle wasting, the circulating IL-6 levels (80–100 ng/ml) were several orders of magnitude higher than the typical levels (0.1–1 ng/ml) measured in cancer cachexia models (Bonetto et al, 2012), thereby preventing it from serving as strong evidence for IL-6's sufficiency in causing cachexia.

Unlike sufficiency, the necessity of IL-6 in causing cachexia in the C26 model appears to be supported by clear evidence. First, studies have shown neutralizing circulating IL-6 with anti-IL-6 antibodies partially prevented body weight loss (Bindels et al, 2018; Strassmann et al, 1992; Yasumoto et al, 1995). Second, as the cancer cells are the main source of circulating IL-6 in the C26 model (Fujiki et al, 1997), inhibiting IL-6 expression in the C26 cells by short-hairpin RNA (shRNA) dramatically reduced circulating IL-6 levels and almost completely prevented body weight loss (Petruzzelli et al, 2014). Based on these results, researchers have deemed IL-6 necessary for causing cachexia in the C26 model. However, surprisingly, there has been no study that completely disrupts the

Department of Molecular Metabolism, Harvard T.H. Chan School of Public Health, Boston, MA, USA. ✉E-mail: shui@hsph.harvard.edu

IL-6 expression by the C26 tumor and determines its effect on the tumor and host. Unlike the antibody treatment and shRNA knockdown experiments described above, such an experiment would create a condition where IL-6 remains at basal levels, thus serving as a more rigorous test of whether IL-6 is necessary for causing cachexia.

In this study, we examined the necessity of IL-6 in cachexia in the C26 model by using CRISPR to knock out the *IL-6* gene in C26 cells and closely monitor tumor growth and body weight. We found that IL-6 KO significantly slowed tumor growth, and while the IL-6 KO tumor eventually grew in size, cachexia still took place, creating a situation where cachexia happens without elevated circulating IL-6. This result directly challenges the view of IL-6 being necessary for causing cachexia in the C26 model. We further demonstrated that IL-6 promotes tumor growth by repressing immune infiltration in tumors. Moreover, using similar approach, we found that leukemia inhibitory factor (LIF), the other proposed cachectic factor in the C26 model, is not necessary for causing cachexia in this model either.

## Results

### Confirmation of the correlation between elevated circulating IL-6 and cachectic phenotypes in the C26 model

To reveal the real role of IL-6 in cachexia in the C26 model, we first aimed to verify the observed strong correlation between circulating IL-6 levels and cachectic phenotypes in this model (Baltgalvis et al, 2008; Pettersen et al, 2017; Strassmann et al, 1992). For this, we employed a well-established cachectic C26 cell line (cxC26) (Bonetto et al, 2016), as well as a non-cachectic C26 cell line (ncxC26) as a control. In this model, cells are subcutaneously injected into syngeneic host CD2F1 mice. To control for differences in growth rates between the two cell lines, we injected half a million ncxC26 cells and one million cxC26 cells, due to faster growth of ncxC26 than cxC26 in the host. Within two weeks post-tumor implantation, mice bearing the cxC26 tumor lost more than 15% body weight, whereas mice bearing the ncxC26 tumor maintained body weight (Fig. 1A), despite similar tumor progression (Fig. 1B). Consistent with the observed body weight loss, lean and fat mass were decreased only in the cxC26 group (Fig. 1C,D). This reduction manifested in smaller white and brown adipose tissues, quadriceps, and heart (Fig. 1E). A decline in muscle function was also observed in the cxC26 group (Fig. 1F). Interestingly, the kidney size was slightly reduced, and the spleen size doubled in cxC26 tumor-bearing mice, whereas liver mass remained unchanged (Fig. 1E).

Having established the model and control, we next measured circulating levels of IL-6 and other cytokines using a 44-cytokine Luminex panel. The measurement was done for six plasma samples collected every other day along the disease progression. The results showed that the circulating level of IL-6 indeed increased over time in the cxC26 group, with significantly higher level already in day 4 and day 6, even proceeding the onset of body weight loss (Fig. 1G). Interestingly, among the only two other significantly elevated cytokines in the panel was IL-11 (Fig. 1G), another member of the IL-6 cytokine family. Note that both IL-6 and IL-11 were excessively released by cxC26 cells under a stress condition

compared to ncxC26 cells (Fig. EV1), suggesting the tumor as an important source for them. To further confirm the elevation of IL-6 in the cxC26 group, we measured circulating IL-6 levels using a highly sensitive immunoassay and found that circulating IL-6 levels were even elevated within 2 days post-tumor implantation (Fig. 1H). Together, these results confirm a strong correlation between circulating IL-6 levels and cachexia in the C26 model.

### Knockout of *IL-6* in the C26 cells delays the progress of cachexia only because it inhibits tumor growth

Next, we aimed to investigate the causality of circulating IL-6 in cancer cachexia. To test whether blocking IL-6 secretion by cachectic tumor prevents body weight loss, we constructed *IL-6* knockout using CRISPR/Cas9 in the cxC26 cells. We first obtained a cxC26 IL-6 knockout pool (IL-6 KOp), which reduced 86% of the secretion of IL-6 in cell culture (Fig. EV2A). To further validate the knockout, we treated our cells with lipopolysaccharide (LPS), which has been shown to induce IL-6 secretion in multiple cell lines (Liu et al, 2017b; Liu et al, 2018; Matsukawa et al, 2021; Xiao et al, 2018). LPS exposure dramatically induced IL-6 secretion in the cxC26 cells, and the induction was diminished by 95% in the IL-6 KOp cells (Fig. EV2A). Inoculated with the IL-6 KOp cells, mice showed a ~2-week delay in body weight loss, consistent with a previous study (Petruzzelli et al, 2014) (Fig. EV2B). Surprisingly, however, the growth of the IL-6 KOp tumor was substantially slower than that of the wild-type cxC26 tumor (Fig. EV2C). In fact, the IL-6 KOp tumors were barely detectable at the stage when the wild-type cxC26 tumors had already caused body weight loss (Fig. EV2C). As this growth defect was not observed in vitro (Fig. EV2D), these results indicate an important role of IL-6 in tumor growth in vivo, raising doubts about its role in cachexia.

To further examine the role of IL-6 in cachexia, we completely disrupted IL-6 secretion by the cxC26 tumor by isolating the two subclones (s1 and s2) of cxC26 IL-6 KO cells (Fig. 2A). Consistent with the IL-6 KOp, IL-6 KO s1 showed severe growth defect. With the injection of one million IL-6 KO s1 cells, the tumor grew slowly in the first week, and then shrunk and disappeared one month post-injection. We then injected ten million cxC26 IL-6 KO s1 cells and after a slow initiation period, the tumor started to grow after 3 weeks (Fig. 2B). Remarkably, with similar tumor size at the endpoint, the cxC26 IL-6 KO s1 tumor caused similar body weight, lean and fat mass loss, muscle mass loss, and muscle function impairment as observed with the wild-type cxC26 tumor (Fig. 2C–K). This is striking because at the endpoint when cachexia occurred, there was no elevation of circulating IL-6 levels in the cxC26 IL-6 KO s1 group compared to the non-tumor group (Fig. 2L), pointing to factors other than IL-6 as mediators of cachexia here. We observed similar outcomes using the cxC26 IL-6 KO s2 cell line (Appendix Fig. S1).

To examine the applicability of these findings in the Balb/c mouse, from which the original C26 cells were derived, we implanted Balb/c mice with one and ten million of cxC26 and cxC26 IL-6 KO s1 cells, respectively. Consistent with our results in the CD2F1 mice, the cxC26 IL-6 KO s1 cells exhibited slower tumor growth and bodyweight loss without an elevation of circulating IL-6 (Appendix Fig. S2). Altogether, these results demonstrate that IL-6 is not necessary for causing cachexia in the C26 model. Instead, the observed prevention of body weight loss upon IL-6 inhibition is simply due to impaired tumor growth in the absence of IL-6.

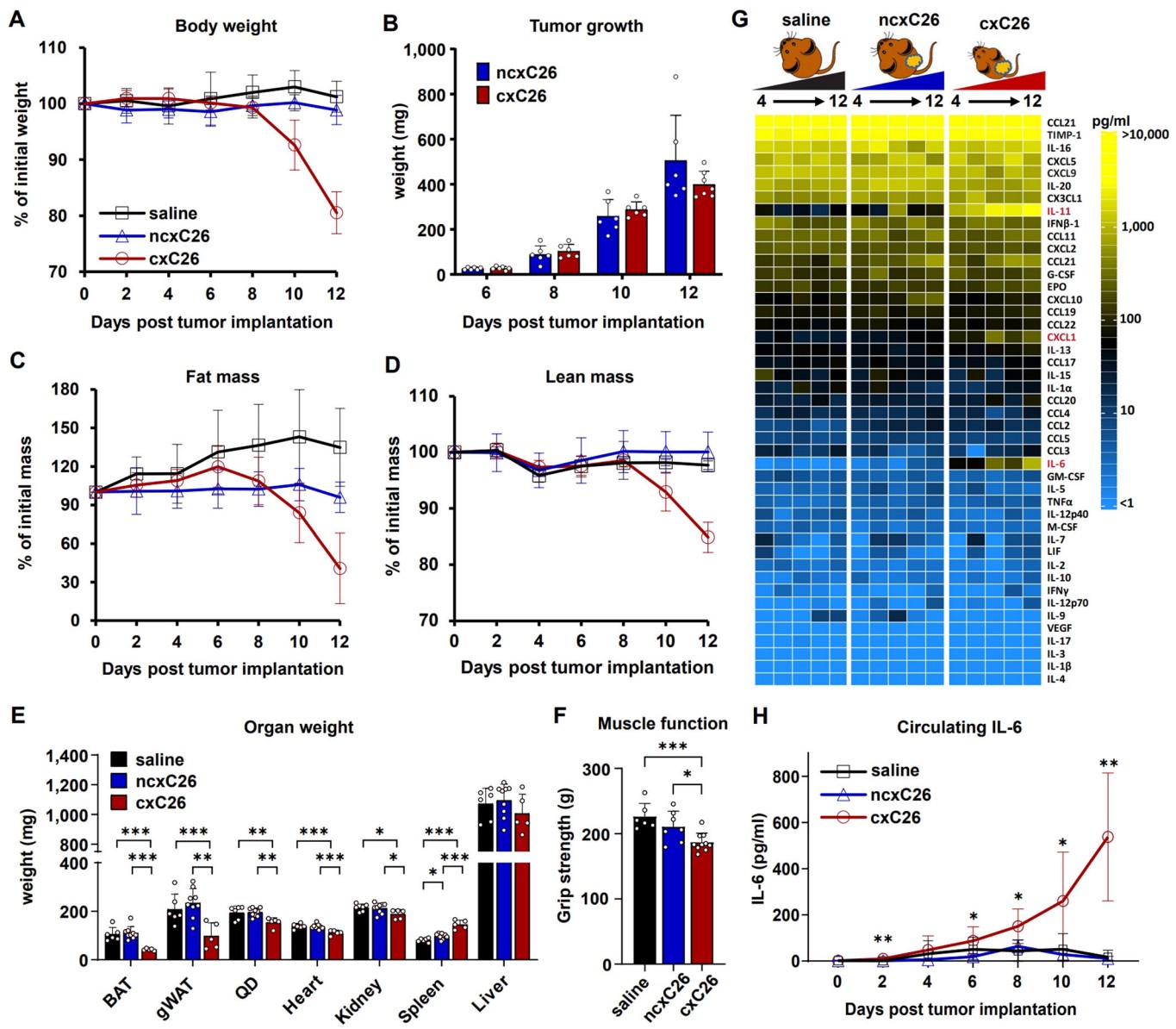

**Figure 1. Elevation of circulating IL-6 in mice bearing the cachectic C26 (cxC26) tumor but not in those bearing the non-cachectic C26 (ncxC26) tumor.**

CD2F1 mice were injected with saline or inoculated with $1 \times 10^6$ cxC26 or $0.5 \times 10^6$ ncxC26 cells. (A) Body weight, (B) tumor mass, (C) fat mass, and (D) lean mass post-implantation. The fat mass and lean mass were measured using EchoMRI. (E) Mass of brown adipose tissue (BAT), gonadal white adipose tissue (gWAT), quadriceps (QD), heart, kidney, spleen, and liver on day 12 post-implantation. (F) Muscle function was evaluated using a grip strength meter. (G) Circulating cytokine profiles were measured for blood samples collected every other day from day 4 to day 12 post-implantation. The cytokine profiles were measured using the Luminex 44-cytokine panel. Cytokines highlighted in red are those with significantly altered levels in cxC26 compared to ncxC26 and saline (adjusted $p$ value <0.01 by two-way ANOVA).
(H) Circulating IL-6 concentration was measured using a Proquantum immunoassay. Data information: (A, C, D) $n = 6$ for saline, $n = 11$ for ncxC26, $n = 12$ for cxC26. (B) $n = 6–7$ for ncxC26 and cxC26. (E) $n = 6$ for saline, $n = 9$ for ncxC26, $n = 5$ for cxC26. (F) $n = 6$ for saline, $n = 7$ for ncxC26, $n = 8$ for cxC26. (G) $n = 5–6$ biologically independent animals per group and time point. (H) $n = 7$ per group. (A–F, H) are shown as the mean ± s.d. Significance of the differences: *$P < 0.05$, **$P < 0.01$, ***$P < 0.001$ between groups by one-way ANOVA (E, F) or two-way ANOVA (H). Source data are available online for this figure.

## Knockout of *LIF* or *IL-11*, another two IL-6 family cytokines, also inhibits tumor growth

Recent studies have implicated LIF as another cachectic factor in the C26 model, based on the key observation that disrupting *LIF* in tumors prevents body weight loss (Arora et al, 2018; Kandarian et al, 2018). Although we did not detect elevated levels of LIF

through the cytokine Luminex panel in our C26 model, potentially due to the very low concentration of circulating LIF (Fig. 1G), we observed a significantly higher secretion of LIF in cxC26 cells compared to ncxC26 cells (Fig. EV1). To assess the causal role of LIF in cachexia in the C26 model, we monitored body weight and tumor growth using an additional cachectic C26 cell line (referred to as C26nci, as it was originally obtained from the National Cancer

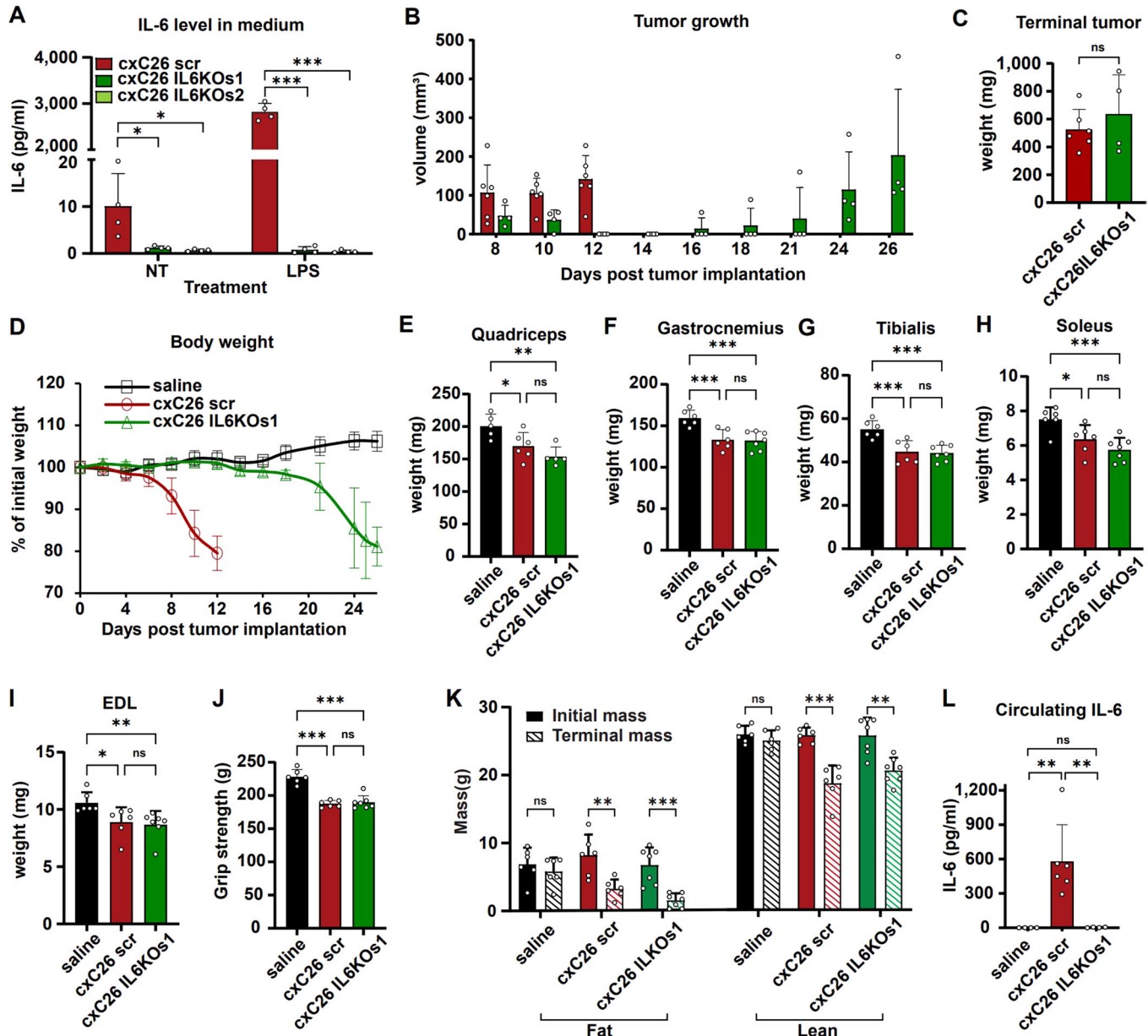

**Figure 2. Disruption of the *IL-6* gene in cachectic C26 cells delays tumor growth, but still leads to body weight loss without elevated circulating IL-6.**

(A) IL-6 levels in conditioned media of cxC26 scr (CRISPR/Cas9 scrambled gRNA control) and cxC26 IL-6 KO subclones 1 (s1) and 2 (s2) with or without lipopolysaccharide (LPS, 2 μg/ml) treatment for 24 h. (B–L) CD2F1 mice were injected with saline, or inoculated with $1 \times 10^6$ cxC26 scr or $1 \times 10^7$ cxC26 IL-6 KO s1 cells. (B) Tumor growth. (C) Tumor weight at the terminal time point. (D) Body weight. Mass of (E) Quadriceps, (F) Gastrocnemius, (G) Tibialis anterior, (H) Soleus, and (I) Extensor digitorum longus (EDL) at the terminal time point. (J) Grip strength at the terminal time point. (K) Fat and lean mass measured with EchoMRI before tumor implantation and at the terminal time point. (L) circulating IL-6 concentration at the terminal time point. Data information: (A) $n = 4$ per group. (B–D, L) $n = 4$ for saline and for cxC26 IL-6 KO s1, $n = 6$ for cxC26. (E) $n = 5$ for saline and cxC26 IL-6 KO s1, $n = 6$ for cxC26. (E) $n = 5$ for saline and cxC26 IL-6 KO s1, $n = 6$ for cxC26. (F–K) $n = 6$ for saline, $n = 6$ for cxC26, and $n = 7$ for cxC26 IL-6 KO s1. All data (A–L) are shown as the mean ± s.d. Significance of the differences: *$P < 0.05$, **$P < 0.01$, ***$P < 0.001$ between groups by one-way ANOVA. ns not significant. Source data are available online for this figure.

Institute) and its LIF KO subclone. A previous study utilized the C26nci cell line and its LIF KO line to demonstrate the cachectic effect of LIF (Kandarian et al, 2018). Consistent with that study, the body weight of the C26nci LIF KO group did not significantly drop, whereas the C26nci group showed a ~20% body weight loss by week 3. However, while monitored for two more weeks, mice

bearing the C26nci LIF KO tumor showed clear body weight loss and muscle wasting, to the same extent as the C26nci group at the endpoint (Fig. 3A,B). Tumor growth data showed that the disruption of *LIF* inhibited tumor growth (Fig. 3C), similar to the effect observed with the *IL-6* disruption. Importantly, the levels of circulating LIF or IL-6 were not elevated in cachectic mice bearing

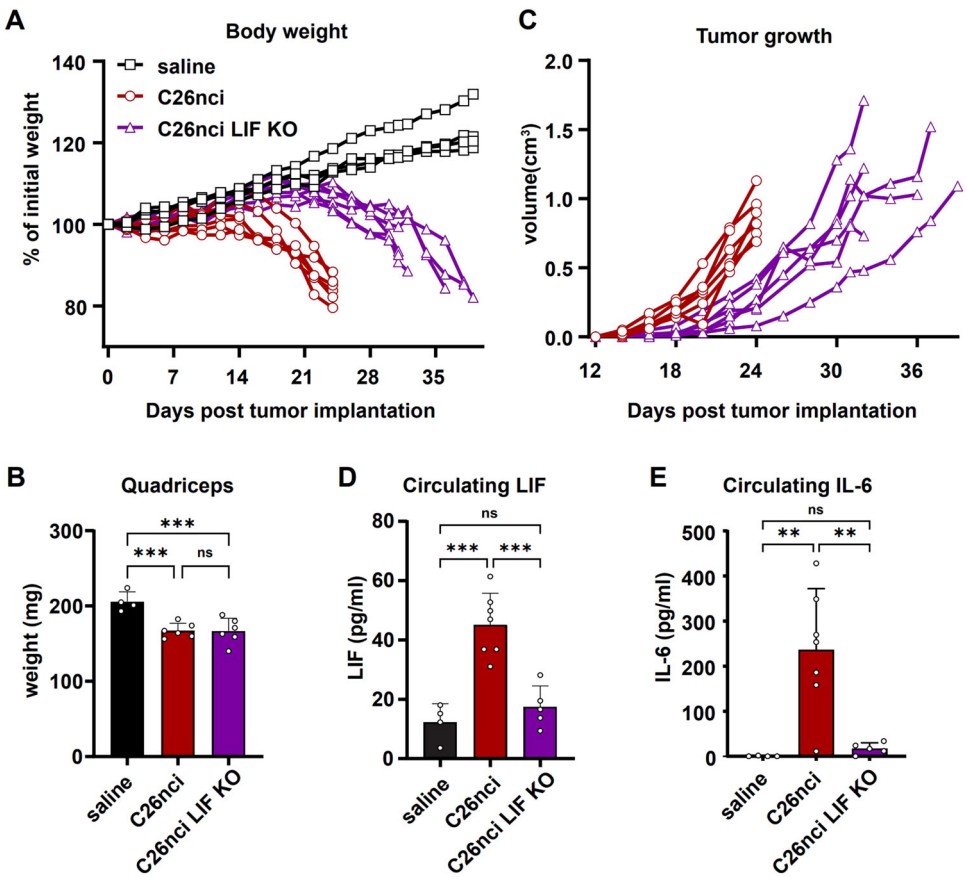

**Figure 3. Disruption of the *LIF* gene in cachectic C26 cells delays tumor growth, but still leads to body weight loss without elevation of circulating LIF or IL-6.**

CD2F1 mice were injected with saline or inoculated with 1 X 10⁶ C26nci or C26nci LIF KO cells. (**A**) Body weight. (**B**) Mass of quadriceps at the terminal time point. (**C**) Tumor growth. (**D**) Concentration of circulating LIF. (**E**) Concentration of circulating IL-6. Data information: (**A–C**) $n = 4$ for saline, $n = 6$ for C26nci and C26nci LIF KO. (**D, E**) $n = 4$ for saline, $n = 7$ for C26nci, $n = 5$ for C26nci LIF KO. (**B, D, E**) are shown as the mean ± s.d. Significance of the differences: **$P < 0.01$, ***$P < 0.001$ between groups by one-way ANOVA. ns not significant. Source data are available online for this figure.

the C26nci LIF KO tumors (Fig. 3D,E). To test whether these results hold for the cxC26 cell line, we constructed a CRISPR pool KO of *LIF* in the cxC26 cells. We observed delayed body weight loss and slowed tumor growth in mice injected with the cxC26 LIF KOp cells (Appendix Fig. S3). Thus, similar to IL-6, LIF appears to mediate cachexia via its effect on tumor growth, and is not necessary for causing cachexia in the C26 model. Moreover, the observation of cachexia without elevated IL-6 levels using another C26 cell line provides further support to our conclusion that IL-6 is not necessary for causing cachexia in the C26 model.

Together with IL-6, circulating IL-11 was revealed by our cytokine data as another cytokine whose level increased over time in the cxC26 group (Fig. 1G). The higher expression of IL-11 was observed in the Lewis lung carcinoma model (Barton and Murphy, 2001), but its role in cancer cachexia was not studied. To investigate the role of IL-11 in the C26 model, we disrupted more than 80% of IL-11 secretion in cxC26 cells using CRISPR/Cas9 (Fig. EV3A). Interestingly, similar to our observations with the IL-6 KO and LIF KO, the ablation of *IL-11* resulted in slowed tumor growth and delayed progression of cachexia (Fig. EV3B–D). Thus, tumor-derived IL-11 is unlikely a cachectic factor in the C26 model.

## Lack of elevation of LIF and IL-11 in mice bearing the cxC26 IL-6 KO s1 tumor

Our knockouts of IL-6, LIF, and IL-11 were single-gene knockouts. It is possible that when one of them is knocked out, the levels of other factors go up in a compensatory fashion to maintain cachexia. To test this, we measured 44 circulating cytokines in the cxC26 scr and cxC26 IL-6 KO s1 tumor-bearing mice when they developed cachexia. The result shows that there was no significant elevation of the measured circulating cytokines in cxC26 IL-6 KO tumor-bearing mice compared to cxC26 scr tumor-bearing mice (Fig. EV4A), indicating a lack of compensatory effects by LIF, IL-11, or any other cytokines in the panel. Consistent with Fig. 2, the cytokine panel measurement further confirmed the lack of elevated circulating IL-6 in cachectic cxC26 IL-6 KO s1 tumor-bearing mice when compared to the saline group (Fig. EV4B). Interestingly, compared to the saline group, neither circulating IL-11 (Fig. EV4C) nor LIF (Fig. EV4D) was elevated in the cxC26 IL-6 KO s1 tumor-bearing mice. The situation of cachexia taking place without elevation of any of the three factors further supports that none of them is necessary for causing cachexia in this model.

In contrast to IL-6, LIF, and IL-11, another proposed cachectic factor, GDF-15 (Kim-Muller et al, 2023; Lerner et al, 2016a; Lerner et al, 2016b), was significantly higher in both cxC26 scr and cxC26 IL-6 KO s1 tumor-bearing mice compared to the saline group (Fig. EV4E), consistent with a role for GDF-15 in cachexia.

Notably, all three cytokines IL-6, LIF, and IL-11 are members of the IL-6 family cytokines, which share a four-helix bundle structure linked by loops and bind to a shared signal-transducing receptor containing the gp130 subunit (Gearing et al, 1992; Rose-John, 2018). Our results indicate their important role in the growth of the cxC26 tumor.

## IL-6 regulates tumor growth via its effects on the host immune system

We next focused on IL-6 and aimed to elucidate its role in tumor growth. IL-6 family cytokines can have autocrine functions, and it is possible that IL-6 directly regulates the tumor growth. For example, IL-6 can act on the JAK/STAT3 pathway, which is aberrantly hyperactivated in various types of tumors, and contribute to tumor growth (Algate et al, 1994; Johnson et al, 2018; Metcalfe et al, 2020; Sreenivasan et al, 2020). To test whether the slowed growth of the IL-6 KO tumors results from disruption of the IL-6 autocrine functions, we disrupted the IL-6 receptor α (IL-6Rα) in the cxC26 cells. In contrast to the cxC26 IL-6 KO tumor, the cxC26 IL-6Rα KOp tumor did not show any difference in tumor growth or body weight loss (Appendix Fig. S4). Thus, the slowed tumor growth of the cxC26 IL-6 KO tumor is independent of the IL-6 autocrine functions.

As described earlier, the injection of one million cxC26 IL-6 KO s1 cells did not lead to tumor growth, and the injection of ten million cxC26 IL-6 KO s1 cells led to a prolonged initiation period before commencing growth (Fig. 2B). These phenomena suggest an active immune response targeting the cancer cells, allowing only a sufficiently large initial population of cancer cells to escape the attack. As it is known that tumors secrete various cytokines to regulate the tumor microenvironment (Baracos et al, 2018), we hypothesized that the cxC26 cells release IL-6 to suppress the host immune response for faster growth. To test this hypothesis, we injected one million of the cxC26 cells and the cxC26 IL-6 KO s1 cells into immunodeficient NOD-SCID mice. Strikingly, the cxC26 IL-6 KO s1 tumors did not show any growth defect compared to the wild-type cxC26 tumor, in contrast to its growth on immunocompetent mice (Fig. 4A,B). Thus, cxC26-derived IL-6 promotes tumor growth by suppressing the host immune response.

Notably, consistent with a previous study (Yasumoto et al, 1995), cachexia developed in the NOD-SCID mice bearing the cxC26 tumor (Fig. 4C–E), suggesting that lymphocytes are not mediators of cachexia. Importantly, cachexia still took place in NOD-SCID mice bearing cxC26 IL-6 KO s1 tumors, despite the absence of elevated circulating IL-6 (Fig. 4F), further supporting that IL-6 is not necessary for causing cachexia in this model.

## Cachectic tumor-derived IL-6 suppresses the infiltration of host immune cells

To further probe the interactions between tumor-derived IL-6 and the host immune response, we evaluated the infiltration of immune cells to the tumor. Using flow cytometry, we quantified various immune cell types with the tumor at two phases: an early phase of tumor growth (day 8) when the cxC26 tumor and the cxC26 IL-6 KO s1 tumor had similar size, and the terminal phase (day 13 for cxC26 and day 30 for cxC26 IL-6 KO s1) when both groups developed the cachexia (Appendix Fig. S5). Strikingly, in the early phase of tumor growth, leukocytes (CD45+) constituted 70% of all cells in the cxC26 IL-6 KO s1 tumor, 3.7-fold more abundant than that in the wild-type cxC26 tumor (Fig. 5). Furthermore, the results revealed a general increase in the infiltration of individual immune cell types within the cxC26 IL-6 KO s1 tumor at the early phase of tumor growth, including B lymphocytes (CD45+/CD19+/CD11b−), macrophages (CD45+/CD11b+/F4/80+/Ly6G−), neutrophils (CD45+/CD11b+/F4/80−/Ly6G+), CD4 + T-lymphocytes (CD45+/CD3+/CD4+/CD8−) and CD8 + T-lymphocytes (CD45+/CD3+/CD4−/CD8+) (Fig. 5). Thus, the cxC26 tumor releases IL-6 to suppress the host immune response by inhibiting the recruitment of cancer-killing immune cells in the tumor.

The immune role of IL-6 was also evident in the RNA-seq data of the cxC26 IL-6 KO s1 tumor. Compared to the cxC26 scr tumor, the significantly changed biological processes were predominantly related to immunity, including leukocyte-mediated cytotoxicity, immune response, and cytokine production (Fig. EV5). Note that consistent with the circulating cytokine measurement (Fig. EV4), the RNA-seq data showed reduced expression of LIF and IL-11 in the IL-6 KO tumor (Appendix Fig. S6).

## Discussion

Identifying the cachectic factors is a major goal of cancer cachexia research. To be classified as such, a factor must meet the criterion that inhibiting it leads to reduced body weight loss without negatively impacting tumor progression. While body weight data is always presented as a central piece of data in studies proposing cachectic factors, it is not uncommon that tumor size and cancer status are inadequately addressed or even ignored. In this study, we subjected one of the widely accepted cachectic factor, IL-6, to this criterion in the C26 model, only to find that it did not meet the criterion as inhibiting IL-6 slowed tumor growth. We then went further and ruled out IL-6 as a cachectic factor in the C26 model by showing that cachexia still took place without elevation of circulating IL-6. Surprisingly, another widely accepted cachectic factor, LIF, was also found to affect cachexia via its effects on tumor growth in the C26 model.

In this study, we focused on the C26 model, in which the role of IL-6 in cancer cachexia was originally proposed. While our results do not draw conclusions regarding other models of cancer cachexia where IL-6 has been implicated as a mediator (Pettersen et al, 2017; Rupert et al, 2021; White et al, 2013), they call for a more thorough examination of IL-6's role in those models. In fact, in studies reporting the preventive effect of cachexia by the disruption of the IL-6 gene in a pancreatic ductal adenocarcinoma model (Rupert et al, 2021), a fibrosarcoma (CHX207) model (Pototschnig et al, 2023) and an Apc^Min/+ colorectal cancer model (Baltgalvis et al, 2008), tumor growth was also reduced. More generally, our results underscore the importance of closely monitoring the impact of a factor on tumor growth in establishing its causality in cachexia.

Despite being widely cited as a cachectic factor, in cancer cachexia patients, circulating IL-6 remained either unchanged (Fortunati et al, 2007; Fujiwara et al, 2014; Lerner et al, 2016a; Ramsey et al, 2019), or showed a modest increase (less than threefold) (Anderson et al, 2022; Costa et al, 2019; Pettersen et al,

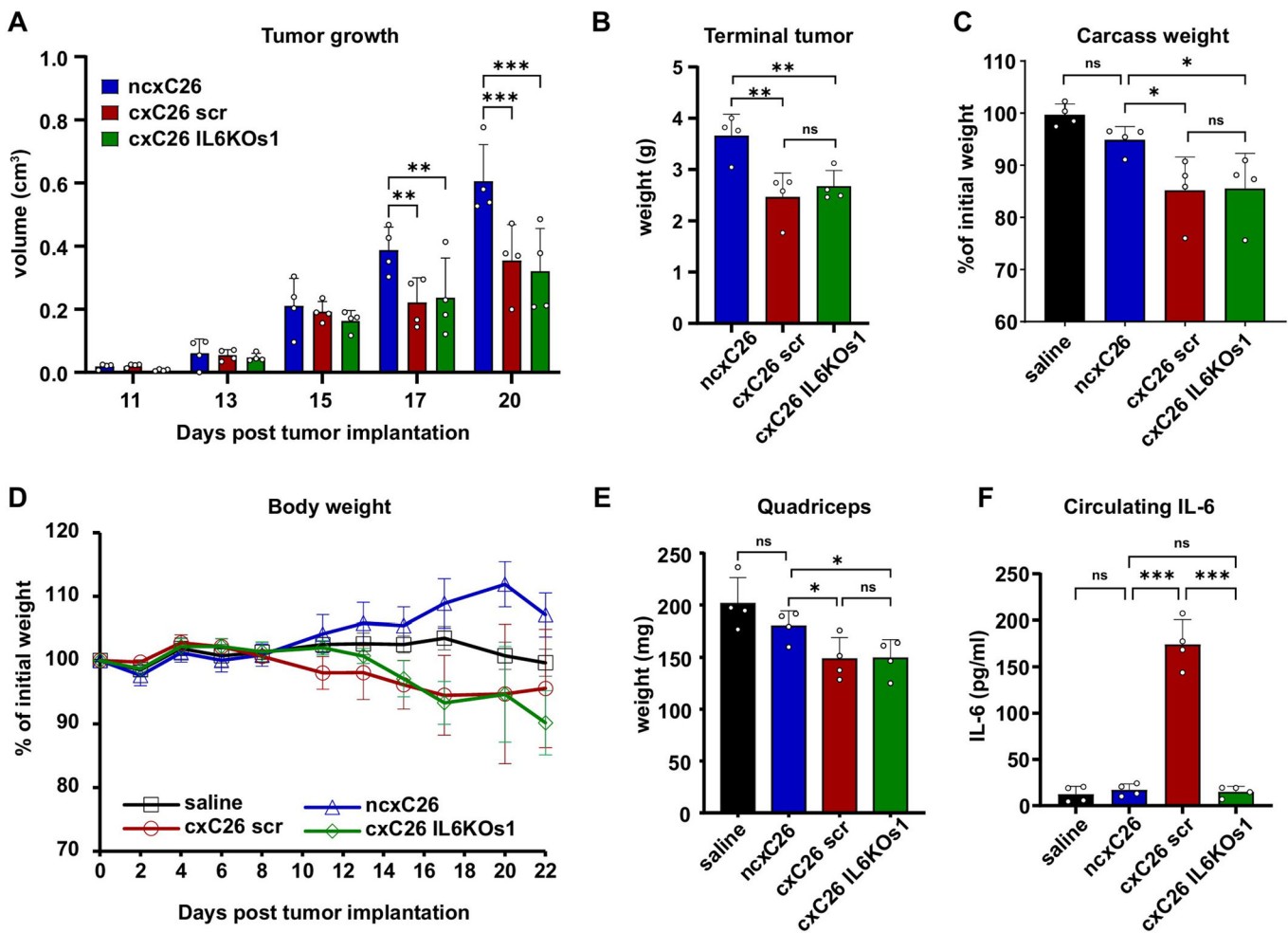

**Figure 4. Deficiency of the host immune system rescues the growth of cxC26 IL-6 KO tumor.**

NOD-SCID mice were inoculated with $1 \times 10^6$ ncxC26, cxC26 scr, or cxC26 IL-6 KO cells. All groups were euthanized simultaneously on day 22. (A) Tumor growth. (B) Terminal tumor weight. (C) Carcass weight at the terminal time point, normalized by the initial body weight. (D) Body weight. (E) Mass of quadriceps, and (F) concentration of circulating IL-6 at the terminal time point. Data information: All data (A–F) are shown as the mean ± s.d. and $n = 4$ per group. Significance of the differences (A–C, E, F): *$P < 0.05$, **$P < 0.01$, ***$P < 0.001$ between groups by one-way ANOVA. ns not significant. Source data are available online for this figure.

2017; Sun et al, 2021), orders of magnitude lower than those measured in mouse models (up to 1000-fold). Moreover, IL-6 is a pleiotropic cytokine, involved not only in immune and inflammatory responses but also in other processes, such as exercise (Kistner et al, 2022; Murakami et al, 2019). For example, in humans, the circulating IL-6 level can be raised ~1000 times in inflammatory states (Waage et al, 1989) and during exercise (Fischer, 2006). Thus, there is no strong data that supports the cachectic role of IL-6 in human cancer patients, in line with our results on IL-6 in the C26 model.

Our results revealed that IL-6 promotes the growth of the cxC26 tumor via its effect on the immune response, which is consistent with the known roles of IL-6 in cancer proliferation (Taher et al, 2018). Directly relevant to our results, IL-6 is known to modulate the host immune system by suppressing the adaptive immune response or recruiting tumor-associated M2 macrophages, which can promote cancer growth and progression (Liu et al, 2017a). In several murine cancer models, disrupting IL-6 has suppressed tumor growth by restoring the anti-tumor activity of the host

immune system (Bent et al, 2021; Narita et al, 2013; Sumida et al, 2012). Specifically in the C26 cancer models, IL-6 level was observed to be negatively correlated with immune cell populations within tumors (Flint et al, 2016), and mice deficient in IL-6 showed increased immune cell infiltration along with reduced tumor growth (Ohno et al, 2017). Thus, our observations of increased infiltration of immune cells (Fig. 5) and slowed tumor growth in the cachectic tumor lacking IL-6 align with the known immunosuppression function of IL-6. However, our unique contributions are that it is the tumor-derived IL-6 that represses immune response and, more importantly, that IL-6's effect on tumor growth has been mistaken as its effect on cachexia.

By undermining the two known cachectic factors in the C26 model, our results call for the search for the real cachectic factors in the model. As demonstrated in our study, knocking out a gene in the cxC26 cells and monitoring the effects on body weight loss and tumor growth is an effective strategy to screen for cachectic genes. With the ncxC26 cell line as a control, the candidate cachectic genes are those genes showing differential expression between the

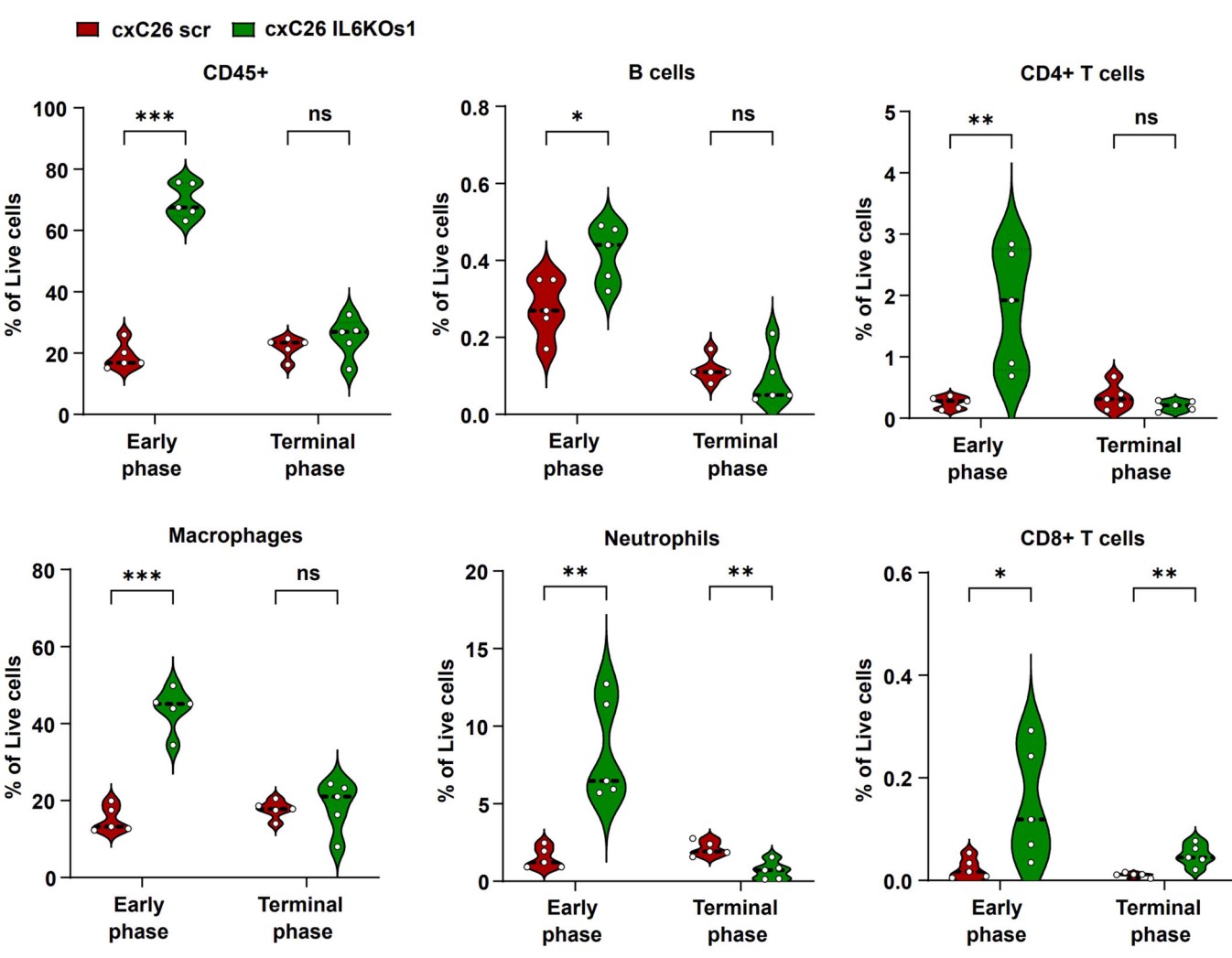

**Figure 5. Infiltration of host immune cells into cxC26 and cxC26 IL-6 KO tumors.**

Mice were inoculated with $1 \times 10^6$ cxC26 scr or $1 \times 10^7$ cxC26 IL-6 KO s1 cells. The tumor-bearing mice were euthanized at the early phase (day 8 for all groups) when they had similar tumor size, and at the terminal phase (day 13 for cxC26 and day 30 for cxC26 IL-6 KO s1) when they developed cachexia. The tumors were dissociated into single-cell suspensions and stained with surface markers. All data were presented as percent of total live cells (7-AAD negative cells). The proportion of CD45+ cells, B cells (CD45+/CD19+/CD11b−), CD4+T-cells (CD45+/CD3+/CD4+/CD8−), CD8+T-cells (CD45+/CD3+/CD4−/CD8+), macrophages (CD45+/CD11b+/F4/80+/Ly6G−), and neutrophils (CD45+/CD11b+/F4/80−/Ly6G+) were enumerated using flow cytometry. Statistical significance (t-test) between groups: *$P < 0.05$, **$P < 0.01$, ***$P < 0.001$. ns not significant. $n = 5$ per group. Source data are available online for this figure.

cxC26 and ncxC26 tumors. The list of candidate genes can be refined by determining whether their corresponding circulating proteins exhibit different levels between the cxC26 and ncxC26 mouse groups. While, in principle, the cachectic factors can be any molecules such as metabolites or lipids, this approach has the potential to reveal tumor-derived proteins that contribute to cachexia.

# Methods

## Animals

Animal care and experimental procedures were conducted with the approval of the Institutional Animal Care and Use Committees (IACUC) of Harvard Medical School and Harvard T.H. Chan School of Public Health. CD2F1, Balb/c, and NOD-SCID male mice were purchased from Charles River and used for this study when the mice were 13–18 weeks old. C26 cells were harvested and implanted in the right flank of mice as previously described (Bonetto et al, 2016). All mice were housed (3–4 per cage) under regular light-dark cycles of 12 h with an ad libitum supply of standard chow diet (PicoLab 5053, LabDiet) and water.

## Cell culture

The cachectic C26 (cxC26) and non-cachectic C26 (ncxC26) cells were kind gifts from Andrea Bonetto (Bonetto et al, 2016) and Nicole Beauchemin, respectively. Nicole Beauchemin obtained the ncxC26 cells from Brattain MG, who established the C26 (also

named CT26 or colon carcinoma 26) cell line (Brattain et al, 1980). The C26nci (C26 from the National Cancer Institute) and C26nci LIF KO were a kind gift from Robert Jackman (Kandarian et al, 2018). All C26 cells were maintained using high glucose DMEM (Corning) with 10% fetal bovine serum (FBS, R&D Systems) and 1% penicillin/streptomycin (P/S, Hyclone).

## Construction of KO cells

To construct the KO cells, two different sets of single-guide RNA (sgRNA) targeting the indicated gene were cloned into the BsmBI site of the lentiCRISPRv2 vector (Addgene #52961) as previously described (Sanjana et al, 2014). The sequences of the sgRNA for each target genes are shown in Appendix Table S1. For lentivirus production, HEK293T cells were plated in DMEM (10% FBS) and transfected with two sets of sgRNA-containing lentiviral vector plasmids along with pMD2.G (Addgene #12259) and psPAX2 (Addgene #12260) using Lipofectamine 3000 (Thermo Fisher Scientific). After 24 h of incubation, the lentivirus-containing medium was collected and filtered through a 0.45 μm syringe filter. The cxC26 cells were then cultured with the lentivirus-containing medium in the presence of polybrene (5 μg/ml, Sigma-Aldrich) for 24 h. Transfected cells were subsequently selected in the presence of puromycin (3 ug/ml, Santa Cruz Biotechnology) in fresh DMEM (+10% FBS and 1% P/S) media. To obtain clonal cell lines of cxC26 IL-6 KO cells, the cxC26 IL-6 KO pool cells were plated onto 96 wells by serial dilution. Two single cells were isolated and expanded in a medium containing puromycin. The two subclones were named as cxC26 IL-6 KO subclone 1 (s1) and cxC26 IL-6 KO subclone 2 (s2), respectively.

## Analysis of cancer cachexia phenotype

To obtain the phenotype of cancer cachexia, body weight, and tumor size were monitored every other day post-tumor implantation. The tumor volume (V) was calculated using the formula $V = (W(2) \times L)/2$, with the length (L) and width (W) of the tumor measured using a caliper. Mice were euthanized when they developed more than 15% body weight loss compared to their initial body weight. To assess body composition, lean mass and fat mass of live mice were measured using an EchoMRI-100 (EchoMRI LLC). As the measurement was done for live mice, the result reflects the composition of the mouse body including the tumor, which due to its small size (<5% of body weight) has minimal impact on the body composition data. To measure grip strength, mice were allowed to grab the metal grid and were then pulled backward by an experimenter until the grasp was released. The peak tension was measured using a Grip Strength meter (Bioseb).

## Cytokines measurement

Circulating cytokines were measured in mouse plasma using the Luminex 44-cytokines panel by Eve Technologies (Calgary, Canada). Circulating LIF and IL-11 levels were also measured using the Mouse LIF Quantikine ELISA Kit (R&D Systems) and IL-11 SimpleStep ELISA kit (abcam), respectively. Circulating IL-6 levels were determined using the highly sensitive IL-6 Mouse ProQuantum qPCR Immunoassay kit (Invitrogen) with real-time PCR.

## GDF-15 measurement

Circulating GDF-15 was measured in mouse plasma using the Mouse/Rat GDF-15 Quantikine ELISA Kit (R&D system) according to the manufacturer's protocol.

## Flow cytometry

CD2F1 mice were inoculated with $1 \times 10^6$ cxC26 cells or $1 \times 10^7$ cxC26 IL-6 KO s1 cells, and the tumors were harvested at day 8 or at the terminal point when tumor-bearing mice developed cachexia. To prepare single-cell suspensions, the harvested tumor was immediately minced using a razor blade in 1 ml of 10% FBS containing RPMI (Gibco), added 2 ml of ice-cold HBSS (Fisher Scientific) containing collagenase I (10 U/ml, Worthington Biochemical), collagenase IV (400 U/ml, Worthington Biochemical), and DNase I (30 U/ml, STEMCELL Technology), and incubated with rotation for 25 min at 37 °C in hybridization oven. Finally, the digests were gently crushed with a pestle and filtered through a 70-μm cell strainer (BD Falcon) to yield single-cell suspensions. To remove red blood cells, the suspension was treated with RBC lysis buffer (BioLegend) on ice for 2 min. The cells were then washed with RPMI media and resuspended in a cell staining buffer (BioLegend). For each sample, $1 \times 10^6$ cells were treated with TruStain FcX^tm Plus (anti-mouse CD16/32, Biolegend) for 10 min and labeled with various fluorescent antibodies. The antibodies used for flow cytometry are listed in Appendix Table S2. Dead cells were excluded by 7-AAD staining (BioLegend). The labeled single cells were analyzed using flow cytometry (BD LSR Fortessa). Flowjo software (Tree Star Inc.) was used for data analysis.

## RNA-seq and data analysis

Fresh tumors were isolated from cxC26 scr and cxC26 IL-6 KO s1 tumor-bearing mice and immediately frozen using liquid nitrogen. RNA extraction, library preparation, sequencing, and analysis were conducted by Azenta Life Sciences (South Plainfield, NJ, USA). Total RNA was extracted from fresh frozen tissue samples using a Qiagen RNeasy Plus Universal mini kit following the manufacturer's instructions (Qiagen, Hilden, Germany). Total RNA was quantified using a Qubit 2.0 Fluorometer (Life Technologies, Carlsbad, CA, USA), and RNA integrity was checked using Agilent TapeStation 4200 (Agilent Technologies, Palo Alto, CA, USA). RNA sequencing libraries were prepared using the NEBNext Ultra RNA Library Prep Kit for Illumina following the manufacturer's instructions (NEB, Ipswich, MA, USA). The samples were sequenced using a $2 \times 150$ bp paired-end (PE) configuration on an Illumina NovaSeq X Plus. Raw RNA-seq data were analyzed on the Galaxy bioinformatics platform (Galaxy, 2022). The raw sequence reads were pre-processed using Trimmomatic and aligned to the mouse genome (mm10) using Hisat2 and summarized by featureCounts, then read counts were normalized using limma-voom. Gene set enrichment analysis was conducted using the package "clusterProfiler" in R software.

## Statistical analysis

All statistical analyses for murine data were performed in GraphPad Prism 9 or R studio. Quantitative data are reported as

mean with standard deviation (s.d.). Two-group comparisons were analyzed by a two-tailed Student's *t*-test and more than two-group comparisons were analyzed by one-way ANOVA or two-way ANOVA. Adjusted *p* values were obtained for multiple comparisons in R using the Benjamini–Hochberg method. For all analyses, a *p* value of <0.05 was considered significant (*$p < 0.05$, **$p < 0.01$, and ***$p < 0.001$). The R package ComplexHeatmap was used to generate the heatmap for the concentration of cytokines.

## Data availability

The RNA-seq data are available through the NCBI SRA repository (accession number: GSE253861).

The source data of this paper are collected in the following database record: biostudies:S-SCDT-10_1038-S44319-024-00144-3.

## Peer review information

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

## Acknowledgements

We would like to thank Dr. Andrea Bonetto (cxC26), Dr. Nicole Beauchemin (ncxC26), and Dr. Robert Jackman (C26nci and C26nci LIF KO) for graciously providing cell lines used for our studies. We are grateful to Dr. Tobias Janowitz, Dr. Eileen White, Dr. Marcus Goncalves, and Dr. Jessalyn Ubellacker for valuable suggestions and critical reading of the manuscript. We thank Nhien Tran for their technical support with the construction of the cxC26 IL-6Rα KO cells. Graphical abstract was created with Biorender. This work was delivered as part of the CANCAN team supported by the Cancer Grand Challenges partnership funded by Cancer Research UK (CGCATF-2021/100034) and the National Cancer Institute (OT2CA2786854). This work was also supported by the National Institute of Diabetes and Digestive and Kidney Diseases (R00DK117066).

## Author contributions

**Young-Yon Kwon**: Conceptualization; Resources; Data curation; Formal analysis; Investigation; Visualization; Methodology; Writing—original draft; Project administration; Writing—review and editing. **Sheng Hui**: Conceptualization; Supervision; Funding acquisition; Investigation; Writing—original draft; Project administration; Writing—review and editing.

Source data underlying figure panels in this paper may have individual authorship assigned. Where available, figure panel/source data authorship is listed in the following database record: biostudies:S-SCDT-10_1038-S44319-024-00144-3.

## Disclosure and competing interests statement

The authors declare no competing interests.

# Expanded View Figures

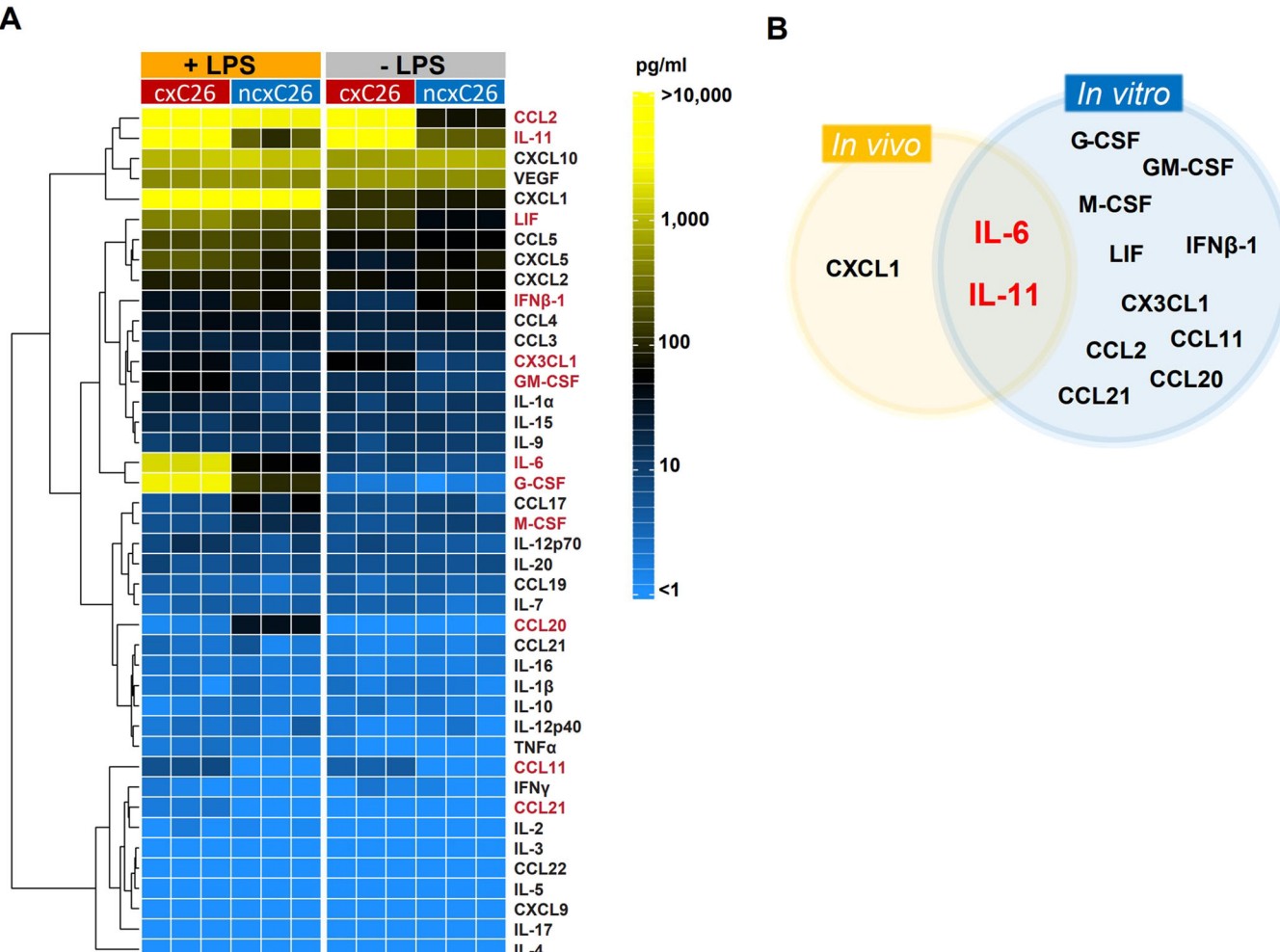

**Figure EV1.  Profiling of secretory cytokines from cxC26 and ncxC26 cells.**

(**A**) Levels of cytokines in conditioned media with or without lipopolysaccharide (LPS, 2 µg/ml) treatment for 24 h. The measurement was done using the Luminex 44-cytokines panel. Cytokines highlighted in red are those with significantly altered levels under LPS treatment between cxC26 and ncxC26 (adjusted *p* value <0.01 by Student's *t*-test, *n* = 3 per group). (**B**) Comparison between significantly changed cytokines by cxC26 in in vivo (Fig. 1G) and in vitro.

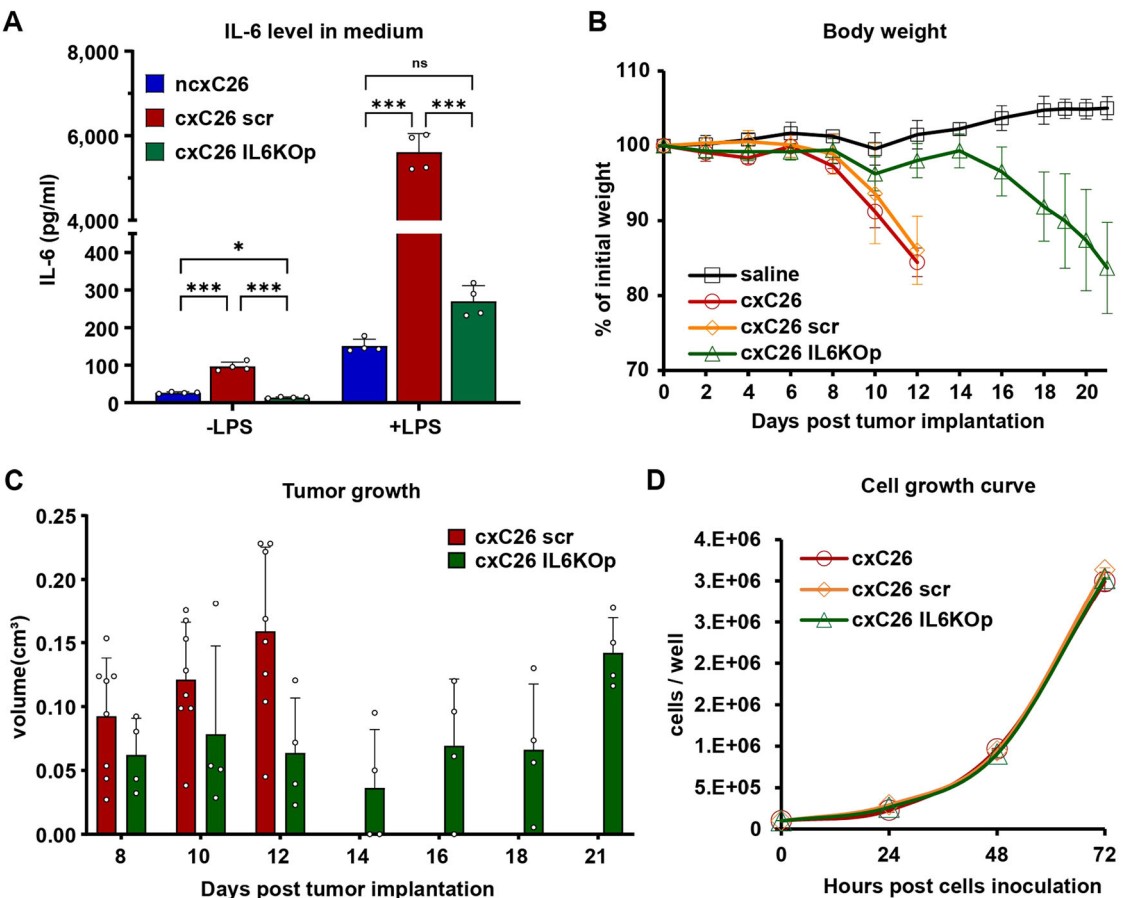

**Figure EV2. Characterization of cxC26 IL-6 KO pool (IL6KOp).**

(A) Levels of IL-6 in conditioned media of cxC26 scr (CRISPR/Cas9 scrambled gRNA control) and cxC26 IL-6 KOp with or without LPS (2 µg/ml) treatment for 24 h. (B) Body weight. (C) Tumor growth. CD2F1 mice were inoculated with $1 \times 10^6$ cxC26, cxC26 scr, or cxC26 IL-6 KOp cells. (D) Growth curves of cxC26, cxC26 scr, and cxC26 IL-6 KOp cells in in vitro. $n = 3$ per group. Data information: (A, B) $n = 4$ for per group. (C) $n = 8$ for cxC26 scr, $n = 4$ for cxC26 IL6KOp (D) $n = 3$ for per group. All data (A–D) are shown as the mean ± s.d. Significance of the differences: *$P < 0.05$, **$P < 0.01$, ***$P < 0.001$ between groups by one-way ANOVA. ns not significant.

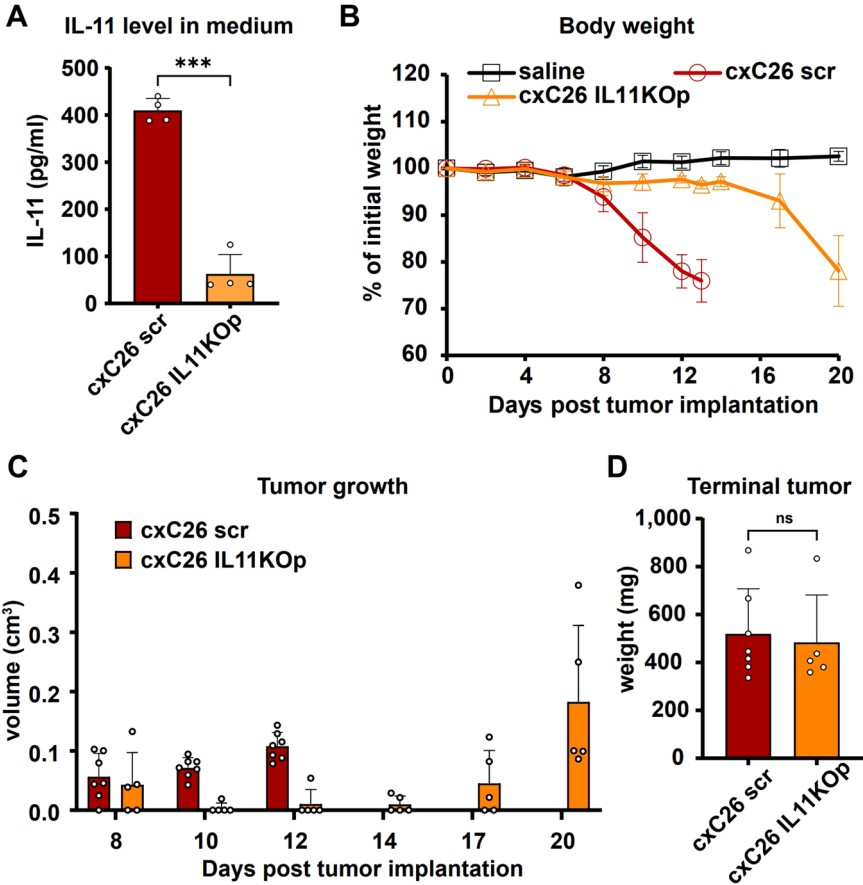

**Figure EV3.  Characterization of *IL-11* knockout in cxC26 cells.**

(A) Levels of IL-11 in conditioned media of cxC26 scr and cxC26 IL-11 knockout pool (KOp). (B) Body weight. (C) Tumor growth. (D) Tumor mass at the terminal time point. CD2F1 mice were injected with saline or inoculated with $1 \times 10^6$ cxC26 scr or cxC26 IL-11 KOp cells. Data information: All data (A–D) are shown as the mean ± s.d. Significance of the differences: ***$P < 0.001$ between groups by Student's $t$-test. ns not significant. $n = 5$ for saline, $n = 7$ for cxC26 scr, $n = 5$ for cxC26 IL-11 KOp.

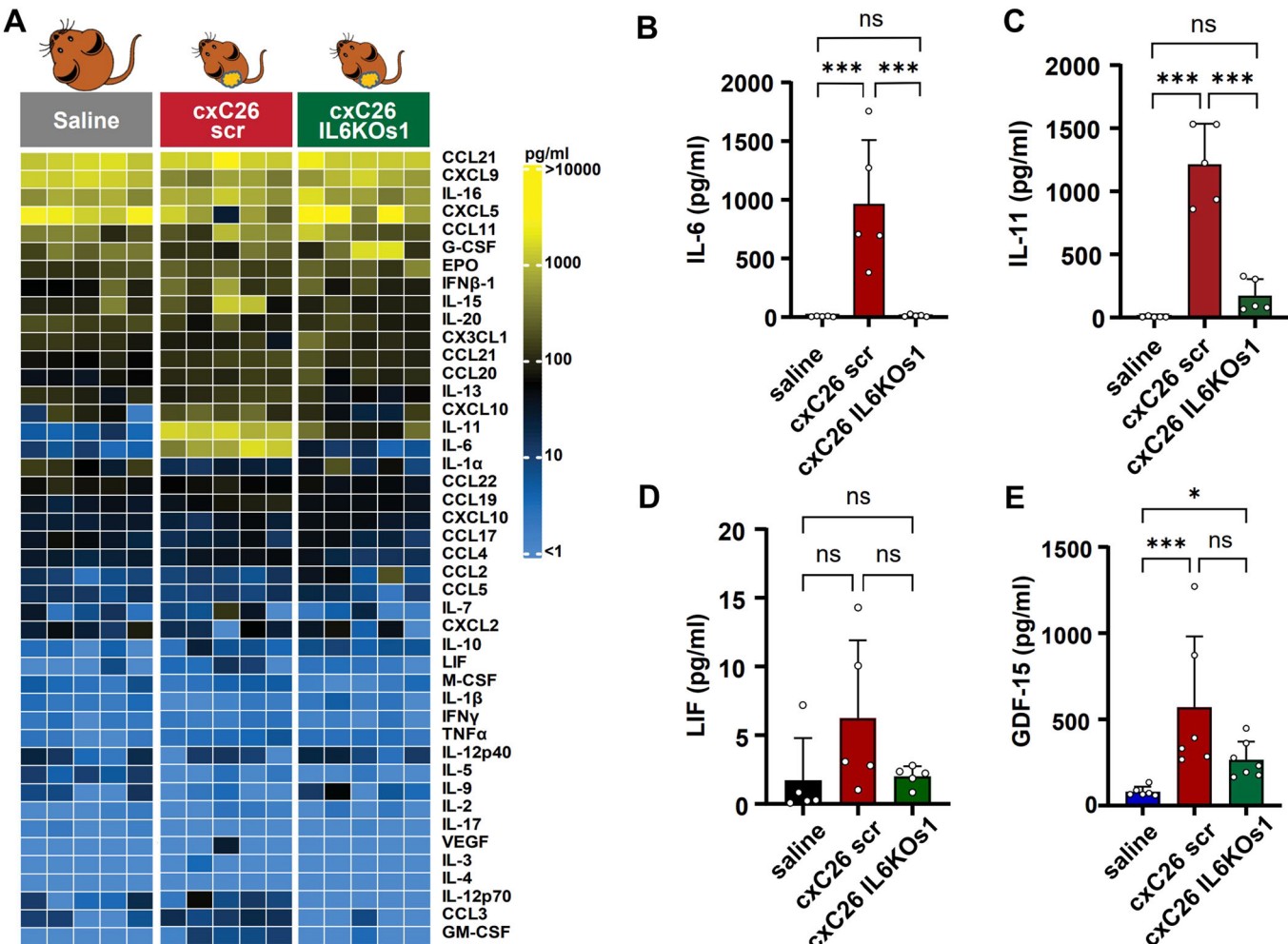

**Figure EV4. Circulating cytokine profile in mice bearing the cxC26 IL-6 KO tumor.**

CD2F1 mice were injected with saline or inoculated with $1 \times 10^6$ cxC26 scr or $1 \times 10^7$ cxC26 IL-6 KO s1 cells. (A) Heatmap for circulating cytokine levels and concentration of circulating (B) IL-6, (C) IL-11, and (D) LIF at the terminal time point. (E) Circulating GDF-15 concentration at the terminal time point. Data information: (A–D) $n = 5$ per group. (E) $n = 6$ for saline and cxC26 scr, $n = 7$ for cxC26 IL-6 KO s1. (B–E) are shown as the mean ± s.d. Significance of the differences: *$P < 0.05$, ***$P < 0.001$ between groups by one-way ANOVA. ns not significant.

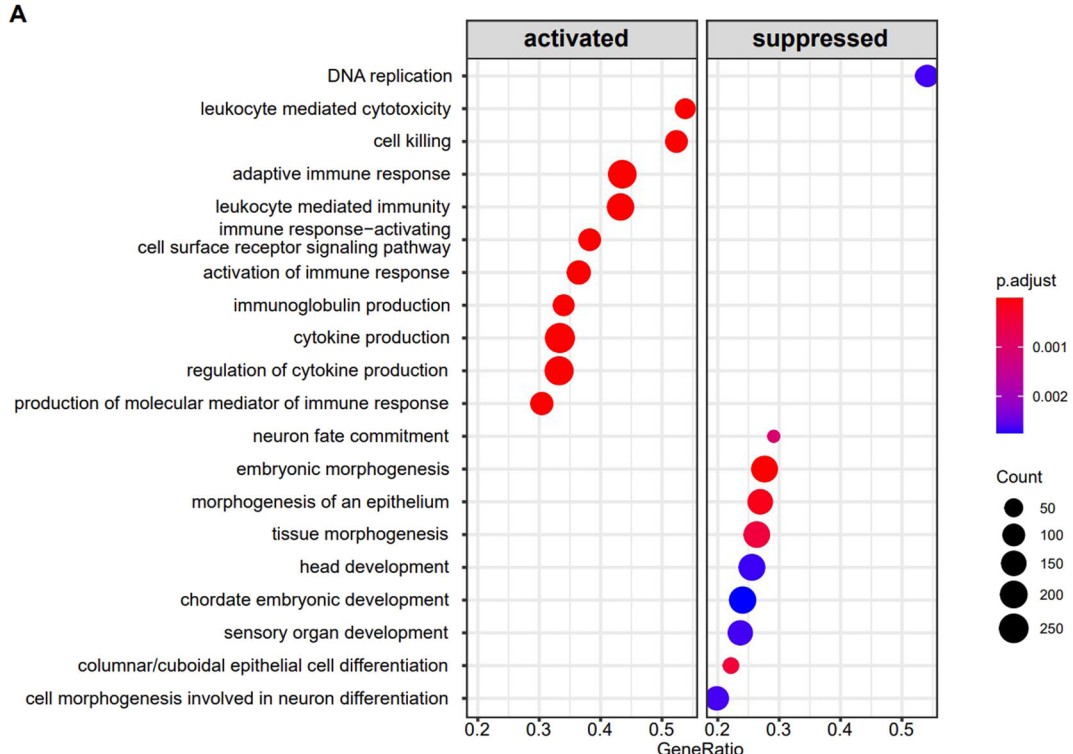

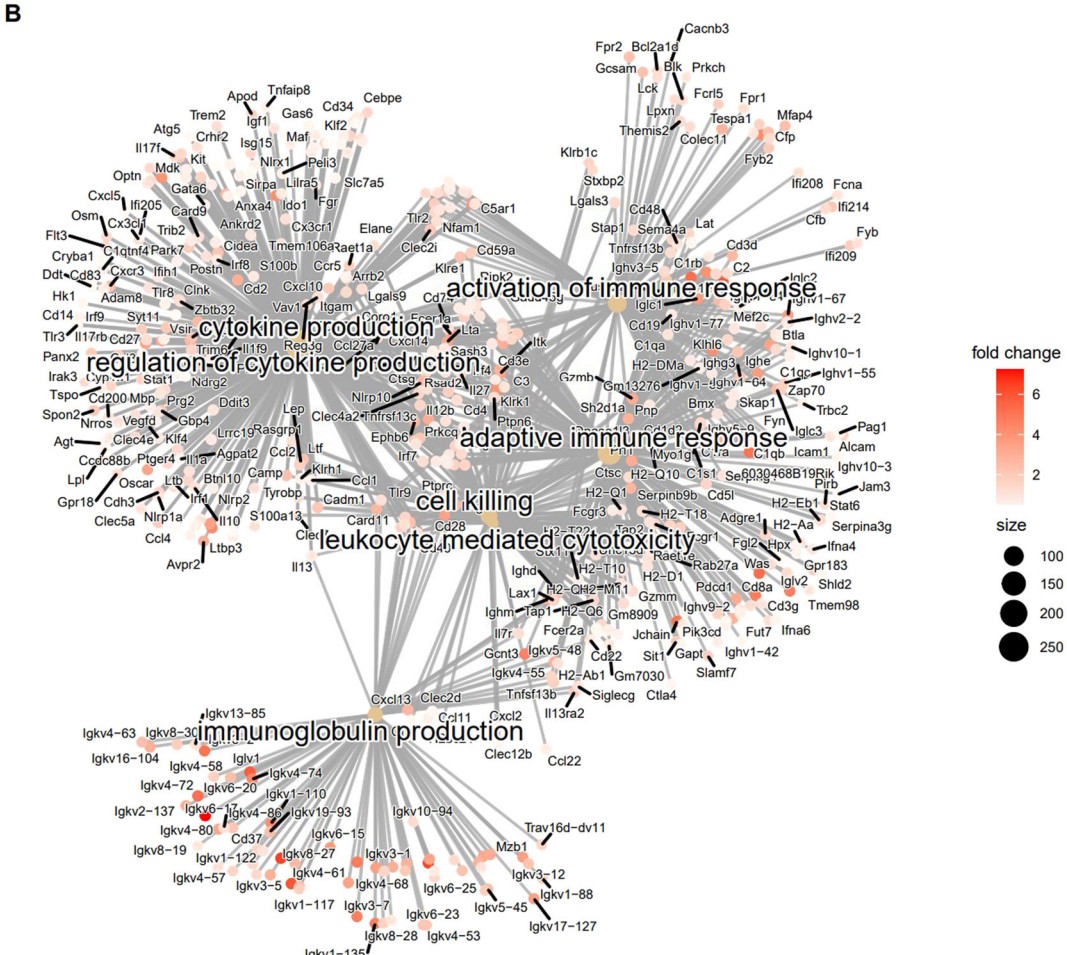

**Figure EV5. Alteration of gene expression by disruption of IL-6 in the cxC26 tumor.**

(A) Elevated and suppressed biological processes in the cxC26 IL-6 KO s1 tumor compared to the cxC26 scr tumor. (B) Network plot of the terms that are the most significantly changed biological processes in the cxC26 IL-6 KO s1 tumor. ClusterProfiler was used for calculating GeneRatio and adjusted $p$ value (FDR) for identifying the significantly changed biological process and drawing the RNA-seq data. $n = 4$ for cxC26 scr and cxC26 IL-6 KO s1.

