## [Peer Review File · EMBO Reports]

IL-6 promotes tumor growth through immune evasion but is dispensable for cachexia

Young-Yon Kwon and Sheng Hui

Corresponding author(s): Sheng Hui (shui@hsph.harvard.edu)

Review Timeline:

Submission Date:	8th Aug 23
Editorial Decision:	13th Sep 23
Revision Received:	24th Jan 24
Editorial Decision:	11th Mar 24
Revision Received:	26th Mar 24
Accepted:	12th Apr 24

Editor: Deniz Senyilmaz Tiebe

Transaction Report:

Dear Dr. Hui,

Thank you for the submission of your research manuscript to our journal, which was now seen by three referees, whose reports are copied below.

The referees express interest in the proposed role of IL-6 in tumor immunity and its dispensability for cachexia. However, they also raise significant concerns that need to be addressed to consider publication here. In particular,

In particular,

- Cachexia needs to be assessed by additional means (referee #1, paragraph 2; referee #3, point 5)
- The tumor identity at different time points should be controlled (referee #2, general comments).
- Whether IL6 loss is compensated by other cachexic cytokines should be tested (referee #3, point 1).
- Food intake should be ruled out from being the cause of weight loss (referee #1, point 3).
- The conclusions regarding JAK/STAT3 need additional support (referee #1, point 5; referee #3, point 2).

Should you be able to address the referee concerns fully, we would like to invite you to submit a revised manuscript. Please revise your manuscript with the understanding that the referee concerns (as in their reports) must be fully addressed and their suggestions taken on board. Please address all referee concerns in a complete point-by-point response. Acceptance of the manuscript will depend on a positive outcome of a second round of review. It is EMBO reports policy to allow a single round of major experimental revision only and acceptance or rejection of the manuscript will therefore depend on the completeness of your responses included in the next, final version of the manuscript.

We realize that it is difficult to revise to a specific deadline. In the interest of protecting the conceptual advance provided by the work, we recommend a revision within 3 months. Please discuss the revision progress ahead of this time with me if you require more time to complete the revisions, or if you have questions or comments regarding the revision (also by video chat).

1. A data availability section providing access to data deposited in public databases is missing (where applicable).
2. Your manuscript contains statistics and error bars based on $n=2$. Please use scatter plots in these cases.

You can submit the revision either as a Scientific Report or as a Research Article. For Scientific Reports, the revised manuscript can contain up to 5 main figures and 5 Expanded View figures, and it should not exceed 27000 characters. If the revision leads to a manuscript with more than 5 main figures it will be published as a Research Article. In this case the Results and Discussion section should be separate. If a Scientific Report is submitted, these sections have to be combined. This will help to shorten the manuscript text by eliminating some redundancy that is inevitable when discussing the same experiments twice. In either case, all materials and methods should be included in the main manuscript file.

<<https://www.embopress.org/page/journal/14693178/authorguide#expandedview>>

4) a .docx formatted letter INCLUDING the reviewers' reports and your detailed point-by-point responses to their comments. As part of the EMBO publication's Transparent Editorial Process, EMBO reports publishes online a Review Process File (RPF) to accompany accepted manuscripts. This File will be published in conjunction with your paper and will include the referee reports, your point-by-point response and all pertinent correspondence relating to the manuscript.

<https://www.embopress.org/page/journal/14693178/authorguide#transparentprocess>

5) a complete author checklist, which you can download from our author guidelines

<https://www.embopress.org/page/journal/14693178/authorguide>. Please insert information in the checklist that is also reflected in the manuscript. The completed author checklist will also be part of the RPF.

6) Please note that all corresponding authors are required to supply an ORCID ID for their name upon submission of a revised manuscript (<<https://orcid.org/>>). Please find instructions on how to link your ORCID ID to your account in our manuscript tracking system in our Author guidelines

<<https://www.embopress.org/page/journal/14693178/authorguide#authorshipguidelines>>

7) Before submitting your revision, primary datasets produced in this study need to be deposited in an appropriate public database (see <https://www.embopress.org/page/journal/14693178/authorguide#datadeposition>). Please remember to provide a reviewer password if the datasets are not yet public. The accession numbers and database should be listed in a formal "Data Availability" section placed after Materials & Method (see also

<https://www.embopress.org/page/journal/14693178/authorguide#datadeposition>). Please note that the Data Availability Section is restricted to new primary data that are part of this study. * Note - All links should resolve to a page where the data can be accessed. *

Additional information on source data and instruction on how to label the files are available:

<https://www.embopress.org/page/journal/14693178/authorguide#sourcedata>

9) Our journal encourages inclusion of *data citations in the reference list* to directly cite datasets that were re-used and obtained from public databases. Data citations in the article text are distinct from normal bibliographical citations and should directly link to the database records from which the data can be accessed. In the main text, data citations are formatted as follows: "Data ref: Smith et al, 2001" or "Data ref: NCBI Sequence Read Archive PRJNA342805, 2017". In the Reference list, data citations must be labeled with "[DATASET]". A data reference must provide the database name, accession number/identifiers and a resolvable link to the landing page from which the data can be accessed at the end of the reference. Further instructions are available at <http://www.embopress.org/page/journal/14693178/authorguide#referencesformat>

10) Regarding data quantification (see Figure Legends:

<https://www.embopress.org/page/journal/14693178/authorguide#figureformat>)

12) Please also note our reference format:

I look forward to seeing a revised version of your manuscript when it is ready. Please let me know if you have questions or comments regarding the revision.

Kind regards,

Deniz Senyilmaz Tiebe

Deniz Senyilmaz Tiebe, PhD
Editor
EMBO Reports

Referee #1:

Kwon and colleagues present an investigation into if three circulating factors that have been proposed to be causative of cancer cachexia are truly required to be released from tumor cells, with a primary emphasis on IL-6. The authors have used CRISPR to delete suspected cachectic factors IL-6, IL-11, and LIF from C-26 tumor cells. The authors attempt to control for variations in rates of tumor growth by varying the number of cells injected, with some success, and make the claim that tumor cell-derived IL-6, IL-11, and LIF are not required for cachexia, as although tumor growth is much slower when one of these factors is deleted. Specifically in the case of IL-6, the authors determine that IL-6 plays a role in shielding tumor cells from the immune system, as IL-6 deficient tumors have much greater numbers of immune cells. The manuscript is likely of broad interest to the cachexia field, and brings a new focus to previous literature that has been underappreciated. However, some claims should be revised or additional data presented to support the claims as currently made.

A significant shortcoming of the manuscript is the use of the mass of only one muscle as an indicator of end-point cachexia, without providing masses of additional muscles or adipose tissue depots, only occasional muscle function data, and no muscle histology. While true that the consensus definition of cancer cachexia focuses solely on muscle, the rapidly expanding literature makes clear that different body compartments are lost at different rates. At current, the authors are missing an opportunity to expand our understanding.

Although the authors attempt to control for the rate of tumor growth by controlling the number of cells injected, this is not always successful and often leads to significant disparities in the length of time the mice with depleted tumors had tumors when compared to parental cell lines. The IL-6KO experiment (Figure 2) clearly shows tumor regression and likely the development of an immune escape of by a resistant clone of cells, and the IL-6KOpool and LIF experiments also have some aspects of tumor regression. Without additional data, it is difficult to determine if weight loss is directly tied to tumor cells themselves, or another aspect of prolonged tumor bearing, such as decreased food or water intake or decreased activity that would have affected mice in the depleted group more severely than the parental group. Another option to further demonstrate that the weight loss seen in IL-6KO cells is not simply due to prolonged tumor-bearing would be pair feeding experiments to demonstrate that decreased food intake over weeks of tumor-bearing is not causative of weight loss.

The timeline of tumor development of C26scr cells across experiments is perplexing to this reviewer. Why would tumor growth would be slower in the NOD-SCID mice for the cxC26scr cells? In Figure 4d, the survival was 22 days, while in Figure 2d, the survival was only 12 days. While true that that there was no difference in the rate of growth of the scrambled and IL-6KO cells in NOD-SCID mice, it's not that the tumor growth of the IL-6KO cells sped up, it's that the IL-6WT cells slowed down. In light of this, the abstract's reference to defective tumor growth of IL-6KO being rescued really isn't accurate.

The claim about JAK/STAT3 independence is overstated, as these were not the disrupted genes - only IL-6Ralpha was disrupted, and therefore the claim should be restricted specifically to this gene.

I think the claim of invalidating two known cachectic factors (line 282) may be unnecessarily strong and a somewhat unfair characterization of what this study adds to the literature. Furthermore, while I agree that deleting genes from tumor cells can be a

useful approach to the study of cachexia, I disagree with the reductionist approach suggested by the authors to identify cachectic factors in immune-deficient mice, particularly given what we now know about how a number of organ systems cross-talk with each other. Human cancer patients have immune systems, and expanding evidence suggests changes in the immune system correlate with cachexia. Softer language should be used to be more fair to the current state of the field.

Minor concerns

The clarity of the manuscript would benefit from editing for grammar and syntax.

While I appreciate the authors nuanced take on sufficiency of IL-6 to cause cachexia versus necessity of IL-6 in cachexia, and I broadly agree with their interpretation of the previous literature, I think that there's some subjectivity to the use of these terms that makes the manuscript more complex for readers to parse than is perhaps necessary. For me, familiar with the same literature that the authors cite and perhaps more deeply engrossed in the human literature, I would have said that IL-6 is not required for cachexia (i.e. cachexia can occur without elevated levels of IL-6) but that it is sufficient for cachexia (due to the neutralizing antibody experiments). I think the manuscript would benefit from considering that readers may view the data presented from this perspective and attempt to revise their introduction with this in mind to aid their readers in finding their perspective.

One caveat to the authors' interpretation of the results of Soda 1994 (line 50) is the length of time that tumor-bearing mice were exposed to elevated circulating levels of IL-6. Yes, the authors saw reduced atrophy with Clone 5, and yes, Clone 5 led to roughly similar circulating IL-6 levels - but the timeline of IL-6 exposure was likely very different in these mice compared to Clone 20.

In line 169, the language should be edited to be more precise that "tumor-derived" LIF is not necessary for cachexia - it is not possible to determine that LIF is not at all required, as animals still had some circulating LIF.

I believe the label for Figure 3b is incorrect and that it is quadriceps mass, not terminal tumor mass.

In line 181, the language surrounding IL-11 as a cachectic factor is not precise - it should relate specifically to tumor-derived IL-11.

Information should be added to the methods to state how the developing tumor was accounted for in longitudinal measures of body composition.

Referee #2:

Summary

The manuscript by Kwon and Hui explores the role of the cytokine IL6 in cancer cachexia. In particular, the authors test the necessity of IL6 action for the development of cachectic phenotypes in mice. To this end, they initially confirmed the correlation between elevated IL6 levels and the severity of cachectic body weight loss in the commonly used C26 transplantation model. By employing CRISPR/cas9-generated C26 IL6 KO cells, they demonstrated that loss of IL6 still promoted (delayed) body weight loss in animals as compared with the parental C26 tumors, associated with a slowed down tumor growth. Similar findings were obtained by inhibiting two other cytokines, LIF and IL11. While the tumor growth inhibition was independent from intra-tumoral IL6-JAK-STAT3 signaling, immunodeficiency in the NOD-SCID model abrogated any growth defects of the IL6 KO tumor cells. Of note, NOD-SCID mice still developed cancer cachexia upon implantation of IL6 KO cells. Flow cytometry showed a significant alteration of different types of immune cells between IL6 wt and KO tumors. Overall, the authors conclude that IL6 suppresses the host immune response to promote tumor growth but is dispensable for cancer cachexia in this model.

General comments

Cachexia remains a major unmet clinical need in a number of severe disease conditions, most notably cancer. Over the past decades, several circulating mediators of cachexia have been identified in a variety of models. However, none of the as-yet identified factors has proven clinical benefit upon therapeutic targeting. In this context, the manuscript by Kwon and Hui addresses an interesting and important issue for the cachexia field, given that the C26 animal model represents the most widely used paradigm in the field and IL6 represents one of the most investigated pathways in the pathogenesis of cancer cachexia. The manuscript is concise and easy to follow. Although the manuscript starts with confirmatory data and lacks a certain mechanistic depth, it still conveys two important messages: a) IL6 is not necessary in the C26 model, and b) investigators should more carefully consider tumor phenotypes when assessing pro- or anti-cachectic factors experimentally. However, one major issue requires further attention by the authors: The IL6 KO tumor cells display a pronounced growth defect, almost disappearing two weeks after animal inoculation and then re-appearing. It is mandatory to clarify to which degree the initial tumor cells (upon inoculation) and the final tumor cells, re-appearing after 3-4 weeks, are still identical, i.e. to rule out any intra-organismal selection process etc. To this end, the authors should run RNA Seq experiments comparing the two tumor cell types and verify and/or document alterations, e.g. do late-stage IL6 KO tumors up- or down-regulate other secreted factors and are

regulated pathways downstream of immune cell stimulation (e.g. using NicheNet analysis) etc. Addition of experimental data will significantly strengthen the case for publication.

Specific comments

- Why was LPS used as a stress inducer in vitro? Does LPS exposure mimic the in vivo situation? Please clarify.
- Fig. 4: How do you explain the difference between the ncxC26 and the cxC26 scr/cxC26 IL6Kos1 in terms of tumor growth? The cxC26 scr also appears to inhibit tumor growth at later stages as compared to ncxC26? Please clarify.

Referee #3:

The present work analyzed the contribution of cancer-dependent IL6 secretion in tumor growth and cachexia onset. The authors used different C26 cancer cells that were genetically modified to block IL6 or LIF expression and tested to ability to growth and to induce body weight decrease and muscle loss in mice. They also used immunocompromised mice to test the contribution of immune response in these processes. They conclude that IL6 is critical to block immune response and to promote cancer growth but is not critical for cachexia onset. While the topic is of great interest and it is crucial to determine the role of IL6/STAT3 pathway in cachexia onset, the conclusions are not supported by the data. The major weakness is the lack of a detailed analyses of the pro-cachectic cytokines in the different C26 cell lines that may compensate the lack of IL6. The authors should consider the following points.

Point1. Figure 2. Authors must check whether compensatory pro-inflammatory or pro-cachectic cytokines are upregulated in C26 IL6 KO cells. In fact, the cytokines detected by the multiplex do not contain all the TGFb members, which are well establish catabolic factors. Moreover, the status of IL11 and LIF, other two inflammatory cytokines that were up in cxC26, were never tested for compensatory upregulation. Authors must check the expression of IL11, LIF, Activin A, GDF11, GDF15, GDF8, TGFb in the serum of the mice implanted with cxC26 IL6KO cell lines and compared with cxC26 scr.

Point2. Figure2. The status of JAK/STAT3 pathway must be tested in muscle of the cxC26 IL6KO versus cxC26 scr tumor bearing mice.

Point3. Figure3 and Figure S6b-d. The concerns in Point1 and 2 also apply to LIF and IL11 KO cell lines.

Point4. A gene expression characterization by RNA seq of the different cxC26 lines should be done to determine how much the different lines are similar or diverge especially in terms of secretory factors.

Point5. The authors often used muscle weight as a readout of muscle loss. However, this is a crude measurement that may be affected by different factors including edema and inflammation. Auhtos should teste fiber size as a better readout of the atrophy process. Moreover, a functional measurement such as grip strength would also important to establish the impact on muscle force production. Indeed, the authors already used this parameter (Fig 1f) to prove the difference between cxC26 versus ncxC26 and therefore, should be used in the other experimental conditions for consistency.

Response to reviewers' comments

We thank all the reviewers for their valuable feedback and suggestions. Please see below our point-to-point responses to the comments. We have made corresponding modifications to the manuscript, with major ones highlighted in red.

Referee #1:

Kwon and colleagues present an investigation into if three circulating factors that have been proposed to be causative of cancer cachexia are truly required to be released from tumor cells, with a primary emphasis on IL-6. The authors have used CRISPR to delete suspected cachectic factors IL-6, IL-11, and LIF from C-26 tumor cells. The authors attempt to control for variations in rates of tumor growth by varying the number of cells injected, with some success, and make the claim that tumor cell-derived IL-6, IL-11, and LIF are not required for cachexia, as although tumor growth is much slower when one of these factors is deleted. Specifically in the case of IL-6, the authors determine that IL-6 plays a role in shielding tumor cells from the immune system, as IL-6 deficient tumors have much greater numbers of immune cells. The manuscript is likely of broad interest to the cachexia field, and brings a new focus to previous literature that has been underappreciated. However, some claims should be revised or additional data presented to support the claims as currently made.

We thank the reviewer for the overall positive assessment of our manuscript.

A significant shortcoming of the manuscript is the use of the mass of only one muscle as an indicator of end-point cachexia, without providing masses of additional muscles or adipose tissue depots, only occasional muscle function data, and no muscle histology. While true that the consensus definition of cancer cachexia focuses solely on muscle, the rapidly expanding literature makes clear that different body compartments are lost at different rates. At current, the authors are missing an opportunity to expand our understanding.

We thank the reviewer for the suggestion. We agree while body weight and muscle mass are important indicators of cachexia, assessment of additional cachectic parameters in mice bearing the IL-6 KO cells will provide stronger support to our claim. As suggested by the reviewer, we have now included the mass of more muscle types (Tibialis, Gastrocnemius, Soleus and EDL), as well as body composition measurement. Moreover, we have included muscle function data (muscle grip strength). These additional data (in the revised Fig. 2) all show the presence of cachexia in mice bearing the cxC26 IL6KO^{s1} cells, providing stronger support to our central claim on the role of IL-6 in cachexia.

Although the authors attempt to control for the rate of tumor growth by controlling the number of cells injected, this is not always successful and often leads to significant disparities in the length of time the mice with depleted tumors had tumors when compared to parental cell lines. The IL-6KO experiment (Figure 2) clearly shows tumor regression and likely the development of an immune escape of by a resistant clone of cells, and the IL-6KO_{pool} and LIF experiments also have some aspects of tumor regression. Without additional data, it is difficult to determine if weight loss is directly tied to tumor cells themselves, or another aspect of prolonged tumor bearing, such as decreased food or water intake or decreased activity that would have affected mice in the depleted group more severely than the parental group. Another option to further demonstrate that the weight loss seen in IL-6KO cells is not simply due to prolonged tumor-bearing would be pair feeding experiments to demonstrate that decreased food intake over weeks of tumor-bearing is not causative of weight loss.

The reviewer raised an interesting question: does longer period of tumor bearing itself lead to cachexia? As far as we know, there is no reported evidence for a positive answer to this question. A non-cachectic tumor such as the nxC26 we used does not lead to cachexia regardless of the length of presence in the host (unless in the extreme situation of huge tumor burden). While setting up the C26 model in our lab, we have tested multiple C26 cell lines from different sources, several of which did not lead to cachexia even after 2 months post tumor implantation. Thus, the length of tumor bearing per se does not appear to be an important parameter contributing to cachexia.

The reviewer was also specifically concerned that longer period of tumor bearing may lead to changed physiology such as reduced food intake (anorexia), explaining the weight loss in the IL-6 KO. We however would like to clarify that anorexia is an important feature of cancer cachexia, in both animal models and cancer patients (Baracos, et al. 2018. Cancer-associated cachexia. Nat Rev Dis Primers). In fact, the most promising cachectic factor now in the field of cancer cachexia, GDF15, is an anorexic factor (Ferrer, *et al.* 2023. Cachexia: A systemic consequence of progressive, unresolved disease. Cell). Specifically in the C26 model that we study, anorexia is known (Flint, T.R., Janowitz, T., et al. 2016. Tumor-induced IL-6 reprograms host metabolism to suppress anti-tumor immunity. Cell metabolism), and for the revision we have measured elevated circulating GDF15 in both mice bearing the WT cxC26 and those bearing the IL-6 KO cells (Fig. EV4). Thus, anorexia is likely still present in mice bearing the KO tumor. But it would not serve as evidence for the existence of cachectic mechanisms that are not related to tumor.

The timeline of tumor development of C26scr cells across experiments is perplexing to this reviewer. Why would tumor growth would be slower in the NOD-SCID mice for the cxC26scr cells? In Figure 4d, the survival was 22 days, while in Figure 2d, the survival was only 12 days. While true that that there was no difference in the rate of growth of the scrambled and IL-6KO cells in NOD-SCID mice, it's not that the tumor growth of the IL-6KO cells sped up, it's that the IL-6WT cells slowed down. In light of this, the abstract's reference to defective tumor growth of IL-6KO being rescued really isn't accurate.

The reviewer is right that the tumor growth of the C26scr and the cachexia progression were different between the NOD-SCID mice and the WT immunocompetent mice. This is because the WT immunocompetent mice were CD2F1 mice (hybrid of BALB/c and DBA/2) while the NOD-SCID mice were in the BALB/c background. (Note that the background for the C26 cells is BALB/c and the cells can grow in both BALB/c and CD2F1 mice.) Thus, the tumor growth and cachexia development cannot be directly compared between mice that have different genetic backgrounds.

The claim about JAK/STAT3 independence is overstated, as these were not the disrupted genes - only IL-6Ralpha was disrupted, and therefore the claim should be restricted specifically to this gene.

We agree with the reviewer that our statement on JAK/STAT3 is overstated as our genetic disruption was only for the IL-6 receptor. We have modified the paragraph and in particular changed the concluding statement from “the slowed growth of the cxC26 IL-6 KO tumor was independent of the IL-6/JAK/STAT3 autocrine signaling pathway” to “the slowed growth of the cxC26 IL-6 KO tumor was independent of the IL-6 autocrine signaling pathways”.

I think the claim of invalidating two known cachectic factors (line 282) may be unnecessarily strong and a somewhat unfair characterization of what this study adds to the literature. Furthermore, while I agree that deleting genes from tumor cells can be a useful approach to the study of cachexia, I disagree with

the reductionist approach suggested by the authors to identify cachectic factors in immune-deficient mice, particularly given what we now know about how a number of organ systems cross-talk with each other. Human cancer patients have immune systems, and expanding evidence suggests changes in the immune system correlate with cachexia. Softer language should be used to be more fair to the current state of the field.

We thank the reviewer for the valuable feedback. We agree on striking the right balance between crediting the literature and adding new findings to it. We have now tuned down our claim by changing “invalidating” to “undermining”. We further agree with the reviewer that dismissing the entire adaptive immune system’s role in cancer cachexia is premature and have now removed this part in the text.

Minor concerns

The clarity of the manuscript would benefit from editing for grammar and syntax.

Thanks for the feedback. For the revision, we have sought help from a professional science editing service to improve the grammar/syntax throughout the text.

While I appreciate the authors nuanced take on sufficiency of IL-6 to cause cachexia versus necessity of IL-6 in cachexia, and I broadly agree with their interpretation of the previous literature, I think that there's some subjectivity to the use of these terms that makes the manuscript more complex for readers to parse than is perhaps necessary. For me, familiar with the same literature that the authors cite and perhaps more deeply engrossed in the human literature, I would have said that IL-6 is not required for cachexia (i.e. cachexia can occur without elevated levels of IL-6) but that it is sufficient for cachexia (due to the neutralizing antibody experiments). I think the manuscript would benefit from considering that readers may view the data presented from this perspective and attempt to revise their introduction with this in mind to aid their readers in finding their perspective.

We understand the reviewer’s concern of our use of “sufficiency” and “necessity”, which may appear as unnecessary jargons to readers. However, as the central goal of our work is to clarify the exact role of IL-6, we feel it is critical to use these well-defined terminologies to evaluate literature results. This is especially necessary for a topic like IL-6, which has been subjected to many studies over the years and is associated with a large literature. Classifying literature evidence into two categories can help prevent confusions. For example, the reviewer mentioned the neutralizing antibody experiments as evidence for IL-6’s sufficiency for cachexia, which however actually demonstrated necessity, as described in our text.

Moreover, as our study focuses on the necessity but not the sufficiency aspect of the factors, it is necessary to resort to these terms for setting up a clear goal of our work.

One caveat to the authors' interpretation of the results of Soda 1994 (line 50) is the length of time that tumor-bearing mice were exposed to elevated circulating levels of IL-6. Yes, the authors saw reduced atrophy with Clone 5, and yes, Clone 5 led to roughly similar circulating IL-6 levels - but the timeline of IL-6 exposure was likely very different in these mice compared to Clone 20.

The reviewer raised an interesting point: the length of time that mice are exposed to elevated IL-6 may impact the cachexia development. This however does not change our interpretation of the results of Soda et al. 1994. As shown in Table III of the paper, Clone 5 at day 21 had similar level of IL-6 as Clone 20 at day 16, suggesting that Clone 5 was likely exposed to longer period of elevated IL-6 at day 21 than

Clone 20 at day 16. As Clone 20 developed clear cachexia at day 16 while Clone 5 did not, the results support that IL-6 is not sufficient for causing cachexia.

In line 169, the language should be edited to be more precise that "tumor-derived" LIF is not necessary for cachexia - it is not possible to determine that LIF is not at all required, as animals still had some circulating LIF.

As the circulating level of LIF was not elevated in the LIF KO (Fig. 3D), the adjective "tumor-derived" is not needed.

I believe the label for Figure 3b is incorrect and that it is quadriceps mass, not terminal tumor mass.

Thanks for catching this mistake. We have now corrected it.

In line 181, the language surrounding IL-11 as a cachectic factor is not precise - it should relate specifically to tumor-derived IL-11.

We agree with the reviewer on this comment as we have not measured the circulating IL-11 level in mice bearing the IL-11 KO tumor. As suggested by the reviewer, we have now added the adjective "tumor-derived" before "IL-11".

Information should be added to the methods to state how the developing tumor was accounted for in longitudinal measures of body composition.

The EchoMRI measurement was done for live mice. The result thus reflects the composition of the animal body including the tumor. This should not be a concern because even the endpoint tumor size is relatively small, <5% of the body weight. In the revision, we have added a sentence on this point in the methods section "Analysis of cancer cachexia phenotype".

Referee #2:

Summary

The manuscript by Kwon and Hui explores the role of the cytokine IL6 in cancer cachexia. In particular, the authors test the necessity of IL6 action for the development of cachectic phenotypes in mice. To this end, they initially confirmed the correlation between elevated IL6 levels and the severity of cachectic body weight loss in the commonly used C26 transplantation model. By employing CRISPR/cas9-generated C26 IL6 KO cells, they demonstrated that loss of IL6 still promoted (delayed) body weight loss in animals as compared with the parental C26 tumors, associated with a slowed down tumor growth. Similar findings were obtained by inhibiting two other cytokines, LIF and IL11. While the tumor growth inhibition was independent from intra-tumoral IL6-JAK-STAT3 signaling, immunodeficiency in the NOD-SCID model abrogated any growth defects of the IL6 KO tumor cells. Of note, NOD-SCID mice still developed cancer cachexia upon implantation of IL6 KO cells. Flow cytometry showed a significant alteration of different types of immune cells between IL6 wt and KO tumors. Overall, the authors conclude that IL6 suppresses the host immune response to promote tumor growth but is dispensable for cancer cachexia in this model.

General comments

Cachexia remains a major unmet clinical need in a number of severe disease conditions, most notably cancer. Over the past decades, several circulating mediators of cachexia have been identified in a variety of models. However, none of the as-yet identified factors has proven clinical benefit upon therapeutic targeting. In this context, the manuscript by Kwon and Hui addresses an interesting and important issue for the cachexia field, given that the C26 animal model represents the most widely used paradigm in the field and IL6 represents one of the most investigated pathways in the pathogenesis of cancer cachexia. The manuscript is concise and easy to follow. Although the manuscript starts with confirmatory data and lacks a certain mechanistic depth, it still conveys two important messages: a) IL6 is not necessary in the C26 model, and b) investigators should more carefully consider tumor phenotypes when assessing pro- or anti-cachectic factors experimentally. However, one major issue requires further attention by the authors: The IL6 KO tumor cells display a pronounced growth defect, almost disappearing two weeks after animal inoculation and then re-appearing. It is mandatory to clarify to which degree the initial tumor cells (upon inoculation) and the final tumor cells, re-appearing after 3-4 weeks, are still identical, i.e. to rule out any intra-organismal selection process etc. To this end, the authors should run RNA Seq experiments comparing the two tumor cell types and verify and/or document alterations, e.g. do late-stage IL6 KO tumors up- or down-regulate other secreted factors and are regulated pathways downstream of immune cell stimulation (e.g. using NicheNet analysis) etc. Addition of experimental data will significantly strengthen the case for publication.

We appreciate the reviewer's accurate summary of our work and the overall positive assessment.

We thank the reviewer for the suggestions to improve our manuscript. As suggested, for the revision, we have performed the RNA-seq experiment on the cxC26 scr and cxC26 IL6KO_{s1} tumors. The gene set enrichment analysis shows that the predominant differences between them are in the immune response pathways (Fig. EV5). Together with our other data on IL-6's role in tumor-host immune interactions, this result supports that the main gene expression differences between the WT and KO tumors are due to the deletion of the IL-6 gene, instead of arising from a selection process. We have made the RNA-seq data publicly available on the NCBI SRA repository [GSE:253861], useful for the community in the effort to identify cachectic factors/pathways.

Further regarding the concern that the cxC26 IL6KO_{s1} tumor might have evolved into a tumor distinct from the WT tumor, we would like to point out that we used clonal KO cells and the only way to change the identity of the tumor is for mutation to happen in one cell and that cell takes over the population. But it would take much longer time than 2-3 weeks to develop a half gram tumor from a single mutated cell.

Specific comments

- Why was LPS used as a stress inducer in vitro? Does LPS exposure mimic the in vivo situation? Please clarify.

The reviewer is right by doubting whether LPS exposure mimics the in vivo situation or not. We do not think so either. We used it because it was shown to induce IL-6 expression in multiple cell lines including CT26 (Xiao et al, 2018), C2C12 (Matsukawa et al, 2021), H292, THP-1 (Liu et al, 2018), and mHypoE-N46 cells (Li et al, 2022). In retrospect, there was no reason to expect that LPS also induces IL-6 in the C26 cells we used just because it induces IL-6 in other cell lines. Nonetheless, the fact that it does induce IL-6 in the C26 cells makes it a useful assay to validate our KO cells. To avoid confusion, now in the revision we have included sentences rationalizing the use of LPS.

- Fig. 4: How do you explain the difference between the ncxC26 and the cxC26 scr/cxC26 IL6Kos1 in terms of tumor growth? The cxC26 scr also appears to inhibit tumor growth at later stages as compared to ncxC26? Please clarify.

We were also puzzled by the difference in growth between the ncxC26 and cxC26 tumors as there was no growth difference in vitro (Fig. EV2D). We hypothesize that this is due to the distinct immune responses to the tumors. I.e., the cxC26 tumor enlists a stronger immune response and despite the immune evasion functions of IL-6 it still grows slower. We further speculate that the reduced food intake in mice bearing the cxC26 tumor may contribute to its reduced growth rate at later stages.

Referee #3:

The present work analyzed the contribution of cancer-dependent IL6 secretion in tumor growth and cachexia onset. The authors used different C26 cancer cells that were genetically modified to block IL6 or LIF expression and tested to ability to growth and to induce body weight decrease and muscle loss in mice. They also used immunocompromised mice to test the contribution of immune response in these processes. They conclude that IL6 is critical to block immune response and to promote cancer growth but is not critical for cachexia onset. While the topic is of great interest and it is crucial to determine the role of IL6/STAT3 pathway in cachexia onset, the conclusions are not supported by the data. The major weakness is the lack of a detailed analyses of the pro-cachectic cytokines in the different C26 cell lines that may compensate the lack of IL6. The authors should consider the following points.

Point1. Figure 2. Authors must check whether compensatory pro-inflammatory or pro-cachectic cytokines are upregulated in C26 IL6 KO cells. In fact, the cytokines detected by the multiplex do not contain all the TGFb members, which are well establish catabolic factors. Moreover, the status of IL11 and LIF, other two inflammatory cytokines that were up in cxC26, were never tested for compensatory upregulation. Authors must check the expression of IL11, LIF, Activin A, GDF11, GDF15, GDF8, TGFb in the serum of the mice implanted with cxC26 IL6KO cell lines and compared with cxC26 scr.

We thank the reviewer for the suggestion. While our main claims on IL-6 and LIF do not depend on whether other cytokines are upregulated or not in the KO cells, we agree that measuring these additional cytokines could provide valuable information on the identity of the cachectic factors. As suggested, for the revision, we have performed the cytokine panel measurement for mice bearing the cxC26 IL6KOs1 tumor (Fig. EV4). The panel covers 44 cytokines including IL-6, LIF, IL-11, and other known cachectic factors. The result shows no compensatory increase of any of these cytokines. Interestingly, neither LIF nor IL-11 level was elevated in mice bearing the IL-6 KO tumor even compared to the non-tumor-bearing mice, offering further support that none of these factors is necessary for cachexia.

While we were not able to cover all the cytokines suggested by the reviewer, we additionally measured GDF15, which has recently emerged as a highly promising cachectic factor (Kim-Muller, *et al.* 2023. GDF15 neutralization restores muscle function and physical performance in a mouse model of cancer cachexia. Cell Reports). Our data shows elevated circulating GDF15 in both mice bearing the cxC26 scr tumor and the cxC26 IL6KOs1 (Fig. EV4E), suggesting GDF15 as a potential cachectic factor in the C26 model. Together with the anorexia phenotype, GDF15 is our research focus in a separate project.

Point2. Figure2. The status of JAK/STAT3 pathway must be tested in muscle of the cxC26 IL6KO versus cxC26 scr tumor bearing mice.

We agree that JAK/STAT3 may play a role in muscle cachexia. Our work however does not make claims on the pathway in muscle. In the original version, we have made statement on the pathway in tumor. Now in the revision, also considering Reviewer 1's comment, we have removed the statement on the role of JAK/STAT3.

Point3. Figure3 and Figure S6b-d. The concerns in Point1 and 2 also apply to LIF and IL11 KO cell lines.

Our new cytokine panel data for mice bearing the IL-6 KO tumor shows a situation where none of the IL-6, LIF and IL-11 was elevated while cachexia took place (Fig. EV4), indicating a lack of compensatory effect between these cytokines. This is further supported by the data in the LIF KO where IL-6 was not elevated (Fig. 3E). While we cannot rule out the possibility that another known cachectic factor is elevated in mice bearing the KO tumors, the behaviors of other factors would not change our central claim that IL-6 or LIF is necessary for causing cachexia.

Point4. A gene expression characterization by RNA seq of the different cxC26 lines should be done to determine how much the different lines are similar or diverge especially in terms of secretory factors.

Thanks for the suggestion. For the revision, we have performed the RNA-seq experiment on the cxC26 scr and cxC26 IL6KO_{s1} tumors. The gene set enrichment analysis shows that the predominant differences between them are in the immune response pathways (Fig. EV5). Together with our other data on IL-6's role in tumor-host immune interactions, this result supports that the main differences between the WT and KO tumors are due to the deletion of the IL-6 gene.

As suggested, we have also performed analysis on secretory factors in the RNA-seq data. Out of the 1723 of genes that encode secreted factors in the RNA-seq data, 52 and 51 were up- and down regulated (FDR < 0.01 and fold change > 2) in cxC26 IL6KO_{s1} tumor compared to cxC26 scr tumor, respectively. In consistent to our GSEA data (Figure EV5), GO analysis of secreted DEGs also indicated the changes of immune related secretory factors as shown below.

Fig. GO analysis of secreted DEGs between cx26 scr and cx26 IL6KO1.

We have made the RNA-seq data publicly available on the NCBI SRA repository [GSE:253861].

Point5. The authors often used muscle weight as a readout of muscle loss. However, this is a crude measurement that may be affected by different factors including edema and inflammation. Auhtos should teste fiber size as a better readout of the atrophy process. Moreover, a functional measurement such as grip strength would also important to establish the impact on muscle force production. Indeed, the authors already used this parameter (Fig 1f) to prove the difference between cx26 versus ncxC26 and therefore, should be used in the other experimental conditions for consistency.

We thank the reviewer for the suggestion. We agree while body weight and muscle mass are important indicators of cachexia, assessment of additional cachectic parameters in mice bearing the IL-6 KO cells will provide stronger support to our claim. We have now included the mass of more muscle types (Tibialis, Gastrocnemius, Soleus and EDL), as well as body composition measurement. While we were not able to obtain the fiber size measurement due to limited time, we have included muscle function data (muscle grip strength). These data are in the revised Fig. 2. These additional data all show the presence of cachexia in mice bearing the cx26 IL6KO1 cells, providing stronger support to our central claim on the role of IL-6 in cachexia.

Dear Dr. Hui,

Thank you for submitting your revised manuscript. It has now been seen by all of the original referees.

As you can see, the referees find that the study is significantly improved during revision and recommend publication. However, I need you to address the points below before I can accept the manuscript.

- Please address the remaining concerns of referee #1 by performing the requested textual changes.
- Please remove the figures from the manuscript text.
- Please rename the 'Competing interests' section as 'Disclosure Statement and Competing Interests'.
- Please remove the Author Contributions section from the manuscript.
- We note the following regarding the figure callouts in the text: Figure 3E is currently not called out in the text. "S" is missing in the names and the callouts of Appendix Tables (the correct callouts should be e.g. Appendix Table S1).
- Please fill in the source data checklist that our Source Data Coordinator Dr. Hannah Sonntag sent on 15.09.2023 (also attached).
- Please make the dataset GSE253861 publicly available and remove the reviewer token from the manuscript.
- Our production/data editors have asked you to clarify several points in the figure legends:
 - o Please note that a separate 'Data Information' section is required in the legends of figures 1a-f, h; 2a-l; 3b, d-e; 4a-f; EV 2a-d; EV 3a-d; EV 4a-e.
 - o Please indicate the statistical test used for data analysis in the legend of figure EV 5a.
 - o Please note that in figures 3b, d-e; EV 4b-e; there is a mismatch between the annotated p values in the figure legend and the annotated p values in the figure file that should be corrected.
- Papers published in EMBO Reports include a 'synopsis' and 'bullet points' to further enhance discoverability. Both are displayed on the html version of the paper and are freely accessible to all readers. The synopsis includes a short standfirst summarizing the study in 1 or 2 sentences (max 35 words) that summarize the paper and are provided by the authors and streamlined by the handling editor. I would therefore ask you to include your synopsis blurb and 3-5 bullet points listing the key experimental findings.
- In addition, please provide an image for the synopsis. This image should provide a rapid overview of the question addressed in the study but still needs to be kept fairly modest since the image size cannot exceed 550 (width) x 300-600 (height) pixels.

Thank you again for giving us to consider your manuscript for EMBO Reports, I look forward to your minor revision.

Kind regards,

Deniz Senyilmaz Tiebe

--

Deniz Senyilmaz Tiebe, PhD
Editor
EMBO Reports

Referee #1:

Thanks to the authors for their thoughtful revision of their manuscript.

While I appreciate the author's assessment that it is difficult to compare tumor growth rates across different strains of mice, both the NOD-SCID mice (figure 4) and the immune competent BALB/c mice (appendix 2) mice have the same background strain, and IL-6ko tumors take approximately 22 days to reach the endpoint tumor mass in both strains of mice. As such, I believe that the reference to defective tumor growth in the abstract remains misleading at best. Instead, the data are perhaps more consistent with promotion of tumor growth by the immune system in mice with competent immune systems.

Line 211, in which the authors state that GDF-15 is elevated and suggests a role for GDF-15 in cachexia, is overstated. Consistent with a role for GDF-15 in cachexia yes, but GDF-15 being elevated does not imply that GDF-15 is causative.

Referee #2:

The authors added relevant data to address the reviewer's comments.

Referee #3:

The authors addressed most of my concerns. The paper is improved

Response to reviewers' comments

Referee #1:

While I appreciate the author's assessment that it is difficult to compare tumor growth rates across different strains of mice, both the NOD-SCID mice (figure 4) and the immune competent BALB/c mice (appendix 2) mice have the same background strain, and IL-6ko tumors take approximately 22 days to reach the endpoint tumor mass in both strains of mice. As such, I believe that the reference to defective tumor growth in the abstract remains misleading at best. Instead, the data are perhaps more consistent with promotion of tumor growth by the immune system in mice with competent immune systems.

The reviewer is right that it is hard to make sense that the IL-6 KO tumor showed similar growth rate between the BALB/c and NOD-SCID mice, despite the same genetic background and hugely different immune systems. Upon more careful thinking and examination of experiments, we realized that the number of injected cells were dramatically different between the two conditions, with 10 million cells in the BALB/c case and 1 million in the NOD-SCID case. The fact that the tumor in NOD-SCID mice reached similar size after 22 days from 1/10 of cells is thus consistent with the antagonist effect of immune system on tumors, clearing up the puzzle. These numbers of inject cells were in fact stated in the figure legends (Appendix Fig. S2 for BALB/c and Fig. 4 for NOD-SCID). We regret that we did not point this out in our previous response.

Line 211, in which the authors state that GDF-15 is elevated and suggests a role for GDF-15 in cachexia, is overstated. Consistent with a role for GDF-15 in cachexia yes, but GDF-15 being elevated does not imply that GDF-15 is causative.

We agree with the reviewer that our data does not support a causative conclusion. We have now changed "suggesting important cachectic roles of GDF-15 in the C26 model" to "consistent with a role for GDF-15 in cachexia".

Dear Tony,

Thank you for submitting your revised manuscript. I have now looked at everything and all is fine. Therefore, I am very pleased to accept your manuscript for publication in EMBO Reports.

Congratulations on a nice work!

Kind regards,

Deniz

--

Deniz Senyilmaz Tiebe, PhD

Editor

EMBO Reports

--
